# Nighttime and Daytime Dark Oxidation Chemistry in Wildfire Plumes: An Observation and Model Analysis of FIREX-AQ Aircraft Data

- 5 Zachary C.J. Decker<sup>1,2,3</sup>, Michael A. Robinson<sup>1,2,3</sup>, Kelley C. Barsanti<sup>4</sup>, Ilann Bourgeois<sup>1,2</sup>, Matthew M. Coggon<sup>1,2</sup>, Joshua P. DiGangi<sup>5</sup>, Glenn S. Diskin<sup>5</sup>, Frank M. Flocke<sup>6</sup>, Alessandro Franchin<sup>1,2,6</sup>, Carley D. Fredrickson<sup>7</sup>, Georgios I. Gkatzelis<sup>1,2,a</sup>, Samuel R. Hall<sup>6</sup>, Hannah Halliday<sup>5,b</sup>, Christopher D. Holmes<sup>8</sup>, L. Gregory Huey<sup>9</sup>, Young Ro Lee<sup>9</sup>, Jakob Lindaas<sup>10</sup>, Ann M. Middlebrook<sup>1</sup>, Denise D. Montzka<sup>6</sup>, Richard Moore<sup>11</sup>, J. Andrew Neuman<sup>1,2</sup>, John B. Nowak<sup>11</sup>, Brett B. Palm<sup>7,c</sup>, Jeff Peischl<sup>1,2</sup>, Felix Piel<sup>12,13</sup>, Pamela
- S. Rickly<sup>1,2</sup>, Andrew W. Rollins<sup>1</sup>, Thomas B. Ryerson<sup>1</sup>, Rebecca H. Schwantes<sup>1,2</sup>, Kanako Sekimoto<sup>14</sup>, Lee Thornhill<sup>5,11</sup>, Joel A. Thornton<sup>7</sup>, Geoffrey S. Tyndall<sup>6</sup>, Kirk Ullmann<sup>6</sup>, Paul Van Rooy<sup>4</sup>, Patrick R. Veres<sup>1</sup>, Carsten Warneke<sup>1,2</sup>, Rebecca A. Washenfelder<sup>1</sup>, Andrew J. Weinheimer<sup>6</sup>, Elizabeth Wiggins<sup>5,15</sup>, Edward Winstead<sup>5,11</sup>, Armin Wisthaler<sup>12,13</sup>, Caroline Womack<sup>1,2</sup>, Steven S. Brown<sup>1,3</sup>

<sup>1</sup>NOAA Chemical Sciences Laboratory (CSL), Boulder, Colorado 80305, USA

15 <sup>2</sup>Cooperative Institute for Research in Environmental Sciences, University of Colorado Boulder, Boulder, Colorado 80309, USA

<sup>3</sup>Department of Chemistry, University of Colorado Boulder, Boulder, Colorado 80309-0215, USA <sup>4</sup>Department of Chemical and Environmental Engineering and College of Engineering – Center for Environmental Research and Technology (CE-CERT), University of California, Riverside, Riverside, CA 92507, USA

- <sup>5</sup>NASA Langley Research Center, MS 483, Hampton, VA 23681, USA
   <sup>6</sup>Atmospheric Chemistry Observations and Modeling Laboratory, National Center for Atmospheric Research, Boulder, CO 80301, USA
   <sup>7</sup>Department of Atmospheric Sciences, University of Washington, Seattle, WA 98195, USA
  - <sup>8</sup>Department of Earth, Ocean, and Atmospheric Science, Florida State University, Tallahassee, FL, 32304, USA
- <sup>9</sup>School of Earth and Atmospheric Sciences, Georgia Institute of Technology, Atlanta, GA 30332, USA <sup>10</sup>Colorado State University, Department of Atmospheric Science, Fort Collins, CO, 80523, USA <sup>11</sup>Science Systems and Applications, Inc. (SSAI), Hampton, VA, 23666, USA <sup>12</sup>Institute for Ion Physics and Applied Physics, University of Innsbruck, 6020 Innsbruck, Austria <sup>13</sup>Department of Chemistry, University of Oslo, 0315 Oslo, Norway
- <sup>14</sup>Graduate School of Nanobioscience, Yokohama City University, Yokohama, Kanagawa, 236-0027, Japan
   <sup>15</sup>Universities Space Research Association, Columbia, MD, USA
   <sup>a</sup>Now at Institute of Energy and Climate Research, IEK-8: Troposphere, Forschungszentrum Jülich GmbH, Jülich, Germany
   <sup>b</sup>Now at EPA Office of Research and Development, RTP, NC 27711, USA
   <sup>c</sup>Now at Atmospheric Chemistry Observations and Modeling Laboratory, National Center for Atmospheric Research, Boulder,
- CO 80301, USA

*Correspondence to*: Steven S. Brown (steven.s.brown@noaa.gov) Abstract.

Wildfires are increasing in size across the western U.S., leading to increases in human smoke exposure and associated negative health impacts. The impact of biomass burning (BB) smoke, including wildfires, on regional air quality depends on emissions, transport, and chemistry, including oxidation of emitted BB volatile organic compounds (BBVOCs) by the hydroxyl radical (OH), nitrate radical (NO<sub>3</sub>), and ozone (O<sub>3</sub>). During the daytime, when light penetrates the plumes, BBVOCs are oxidized mainly by  $O_3$  and OH. In contrast, at night, or in optically dense plumes, BBVOCs are oxidized mainly by  $O_3$  and NO<sub>3</sub>. This

- work focuses on the transition between daytime and nighttime oxidation, which has significant implications for the formation of secondary pollutants and loss of nitrogen oxides ( $NO_x = NO + NO_2$ ), and has been understudied. We present wildfire plume observations made during FIREX-AQ (Fire Influence on Regional to Global Environments and Air Quality), a field campaign involving multiple aircraft, ground, satellite, and mobile platforms that took place in the United States in the summer of 2019 to study both wildfire and agricultural burning emissions and atmospheric chemistry. We use observations from two research
- aircraft, the NASA DC-8 and the NOAA Twin Otter, with a detailed chemical box model, including updated phenolic mechanisms, to analyze smoke sampled during mid-day, sunset, and nighttime. Aircraft observations suggest a range of NO<sub>3</sub> production rates  $(0.1 1.5 \text{ ppbv h}^{-1})$  in plumes transported both mid-day and after dark. Modeled initial instantaneous reactivity toward BBVOCs for NO<sub>3</sub>, OH, and O<sub>3</sub> is 80.1 %, 87.7 %, 99.6 %, respectively. Initial NO<sub>3</sub> reactivity is  $10 10^4$  times greater than typical values in forested or urban environments and reactions with BBVOCs account for > 97 % of NO<sub>3</sub> loss in sunlit
- plumes (jNO<sub>2</sub> up to  $4 \times 10^{-3}$  s<sup>-1</sup>), while conventional photochemical NO<sub>3</sub> loss through reaction with NO and photolysis are minor pathways. Alkenes and furans are mostly oxidized by OH and O<sub>3</sub> (11 – 43%, 54 – 88% for alkenes; 18 – 55%, 39 – 76%, for furans, respectively), but phenolic oxidation is split between NO<sub>3</sub>, O<sub>3</sub>, and OH (26 – 52%, 22 – 43%, 16 – 33%, respectively). Nitrate radical oxidation accounts for 26 – 52% of phenolic chemical loss in sunset plumes and in an optically thick plume. Nitrocatechol yields varied between 33% and 45%, and NO<sub>3</sub> chemistry in BB plumes emitted late in the day is
- responsible for 72 92 % (84 % in an optically thick mid-day plume) of nitrocatechol formation and controls nitrophenolic formation overall. As a result, overnight nitrophenolic formation pathways account for  $56 \pm 2$  % of NO<sub>x</sub> loss by sunrise the following day. In all but one overnight plume we model, there is remaining NO<sub>x</sub> (13 % 57 %) and BBVOCs (8 % 72 %) at sunrise.

#### **1** Introduction

- It is well known that biomass burning (BB), including wildfires, can have large impacts on air quality at local, regional and global scales (Jaffe et al., 2020). The relative impact and importance of wildfire smoke on air quality in the western U.S. is increasing with decreasing anthropogenic volatile organic compound (VOC) and nitrogen oxide ( $NO_x = NO + NO_2$ ) emissions (Bishop and Haugen, 2018; Silvern et al., 2019; Warneke et al., 2012; Xing et al., 2015). This increase is compounded by growing wildfire emissions caused by anthropogenic influences such as human-caused climate change and past wildland
- management practices. Twentieth century suppression of western U.S. wildfires has led to increased fuel loadings and thus fire potential (Higuera et al., 2015; Marlon et al., 2012; Parks et al., 2015). A warmer and drier climate in the western U.S. resulting from human-caused climate change has exacerbated fire potential and has resulted in an increase in the frequency of large wildfires since the 1980s (Abatzoglou and Williams, 2016; Balch et al., 2017; Barbero et al., 2015; Dennison et al., 2014; Marlon et al., 2012; Westerling et al., 2006; Westerling, 2016; Williams et al., 2019).

- Wildfires emit NO<sub>x</sub>, nitrous acid (HONO), biomass burning VOCs (BBVOCs) and particulate matter (PM) that evolve chemically on a range of time scales, from seconds to weeks downwind (Akagi et al., 2011; Andreae and Merlet, 2001; Decker et al., 2019; Hatch et al., 2015, 2017; Koss et al., 2018; Palm et al., 2020). These emissions and their chemical products influence air quality through ozone (O<sub>3</sub>) production, emitted PM, and secondary organic aerosol formation (SOA) (Brey et al., 2018; Jaffe et al., 2020; Jaffe and Wigder, 2012; Lu et al., 2016; Palm et al., 2020; Phuleria et al., 2005). However, the evolution
- of the smoke downwind is influenced by several variables such as fuel type, burn conditions, moisture content, nitrogen content, meteorology, and time of day.

Like most atmospheric oxidation processes, the oxidation of BBVOCs is influenced by three key atmospheric oxidants:  $O_3$ , the hydroxyl radical (OH), and the nitrate radical (NO<sub>3</sub>). The amount of each oxidant present in a plume is influenced by emissions of NO<sub>x</sub>, plume mixing with background air, and the amount of sunlight that penetrates a plume. Photolysis of HONO

can be an important source of  $HO_x$  (=  $OH + HO_2$ ) in the first three hours of aging for wildfires sampled in the western U.S. (Peng et al., 2020). Further, atmospheric background levels of O<sub>3</sub>, as well as photochemical O<sub>3</sub> production within a smoke plume, can provide O<sub>3</sub> for plume oxidation (Jaffe and Wigder, 2012). However, there is limited understanding of the role of NO<sub>3</sub> oxidation in biomass burning plumes.

During daytime, NO<sub>3</sub> is rapidly destroyed by photolysis (R1), and in urban plumes it is destroyed even more rapidly by reaction with NO (R2,  $\tau$  <10 s) (Brown and Stutz, 2012; Wayne et al., 1991).

$$NO_3 + hv \rightarrow NO_2 + O$$
 (R1)

$$NO_3 + NO \rightarrow 2NO_2 \tag{R2}$$

Therefore, although the role of  $NO_3$  in nighttime BBVOC oxidation has been considered previously, the role of  $NO_3$  as a daytime oxidant has been neglected (Decker et al., 2019; Keywood et al., 2015; Kodros et al., 2020; Palm et al., 2020).

Despite the potential for rapid loss of NO<sub>3</sub> with sunlight and NO, wildfire plumes provide a unique environment which promotes NO<sub>3</sub> chemistry. NO<sub>3</sub> is produced within a smoke plume by the gas-phase reaction of O<sub>3</sub> and NO<sub>2</sub> (R3) and is a precursor for N<sub>2</sub>O<sub>5</sub> (R4), a NO<sub>x</sub> reservoir (Brown and Stutz, 2012). N<sub>2</sub>O<sub>5</sub> may undergo heterogeneous uptake to form ClNO<sub>2</sub> and HNO<sub>3</sub> according to the branching ratio  $\phi$  (R5) (Chang et al., 2011; McDuffie et al., 2018). NO<sub>3</sub> can also be directly taken up by aerosol (R6) or react with BBVOCs (R7).

$$100 \quad \mathrm{NO}_2 + \mathrm{O}_3 \to \mathrm{NO}_3 + \mathrm{O}_2 \tag{R3}$$

$$NO_3 + NO_2 \rightleftharpoons N_2O_5$$
 (R4)

$$N_2O_{5(g)} + \operatorname{aerosol} \rightarrow \phi \text{ClNO}_2 + (2 - \phi)\text{HNO}_3$$
(R5)

$$NO_3 + aerosol \rightarrow products$$
 (R6)

(R7)

$$NO_3 + BBVOCs \rightarrow products$$

Modeled NO<sub>3</sub> reactivity was found to be mostly (>99 %) from reactions with BBVOCs (R7) as opposed to heterogeneous reactions with aerosol particles (R5 – 6) in an agricultural burning plume sampled after sunset (Decker et al., 2019). This is the result of elevated concentrations of several highly reactive BBVOCs within the plume. Specifically, directly emitted aromatic alcohols (phenolics, i.e. 6-membered aromatic rings with an alcohol functional group, which are distinct from the broader class of oxygenated aromatics that also includes furans, furfuals, etc.) react with NO3 at near the gas-kinetic limit to

- form nitrophenolics, a subset of nitroaromatics, and secondary organic aerosol (Finewax et al., 2018; Lauraguais et al., 2014; Liu et al., 2019; Xie et al., 2017). Nitrophenolics absorb strongly in the ultraviolet and visible regions of the solar spectrum, and are expected to significantly contribute to BrC absorption (Palm et al., 2020; Selimovic et al., 2020). Phenolic reactions with OH in the presence of  $NO_x$  also form nitrophenolics, but at one third the yield (Finewax et al., 2018).
- Wildfire emissions typically peak in the mid-afternoon to evening, and continue to emit smoke into the night (Giglio, 2007;
  Li et al., 2019). Furthermore, large smoke plumes can be optically thick, with little photolysis at their center. This means that most smoke plumes will be oxidized in the dark during some, if not all, of their transport. Yet, the vast majority of in-situ field investigations of biomass burning smoke has been conducted under sunlight, and most analyses of daytime smoke plumes have so far focused on plume oxidation by OH and O<sub>3</sub> only (Coggon et al., 2019; Keywood et al., 2015; Liu et al., 2016; Palm et al., 2020).
- In the summer of 2019, both the NOAA Twin Otter and the NASA DC-8 aircraft executed a series of research flights sampling smoke plumes as part of the Fire Influence on Regional to Global Environments and Air Quality (FIREX-AQ) campaign. Here, we present a detailed analysis of smoke plumes from three fires using observations from FIREX-AQ to constrain a detailed zero-dimensional (0-D) chemical box model. We investigate one optically thick plume emitted mid-day, three smoke plumes emitted near or at sunset, and one theoretical plume emitted after sunset. We discuss the reactivity and competitive oxidation
- for all oxidants, NO<sub>3</sub>, O<sub>3</sub>, and OH, toward a suite of BBVOCs. Further, we detail the oxidation pathways of phenolics, discuss the variables that affect the yield of nitrophenolics, and describe how nitrophenolics have a significant impact on  $NO_x$  loss and fate.

#### 2 Methods

#### **2.1 Aircraft Measurements**

- FIREX-AQ was a large-scale multi-platform campaign that took place during the summer of 2019 in the United States to study both wildfire and agricultural burning smoke. Both the NOAA Twin Otter and the NASA DC-8 aircraft executed a series of research flights sampling smoke plumes as part of this campaign. A main science goal of the NOAA Twin Otter was to investigate nighttime plume chemistry. However, due to a less active fire season in 2019 (NIFC, 2019) and to the decreasing smoke injection height with time of day for the sampled fires, smoke emitted after dark proved difficult to sample reliably
- within the altitude range of the NOAA Twin Otter. While the NOAA Twin Otter sampled over a dozen plumes after sunset, plume age estimates (described below) suggest these plumes were emitted before or at sunset. The NASA DC-8 aircraft sampled large, optically thick, plumes both mid-day and near sunset. In the following sections we briefly describe the instrumentation used for this analysis, which are listed in SI Table 1. More information and data can be found at https://csl.noaa.gov/projects/firex-aq/twinotterCHEM/, https://espo.nasa.gov/firex-aq, and https://www-
- air.larc.nasa.gov/missions/firex-aq/index.html.

#### 2.1.1 NOAA Twin Otter Instrument Descriptions

The NOAA Twin Otter sampled nine wildfires with 39 flights between 3 August 2019 and 5 September 2019 in the western U.S. The aircraft was based mainly in Boise, ID and briefly in Cedar City, UT. The NOAA Twin Otter payload limited flight duration to 3.0 h or less and the aircraft typically flew 2 - 3 times in a day to achieve plume sampling from mid-afternoon into

- the night. Aircraft speed was  $71.8 \pm 3.8 \text{ m s}^{-1}$  (average  $\pm 1 \cdot \sigma$ ), which yields a horizontal resolution of ~70 m for the in situ 1 s measurements. Attempts to probe the same airmass downwind, known as Lagrangian sampling, proved difficult to achieve due to complex plume structure, terrain and airspace. Therefore, we define the sampling strategy as semi-Lagrangian. Even so, estimated emission times (calculated from estimated plume ages) suggest smoke sampled on successive intercepts at the Castle and Cow plume centers were emitted within 3- and 10-min time periods, respectively. However, plume age uncertainties
- for the Cow plume are large (SI Table 2).

This analysis uses NOAA Twin Otter observations of BBVOCs and HONO from a University of Washington Iodide High Resolution Time of Flight Chemical Ionization Mass Spectrometer (UW I<sup>-</sup> HR ToF CIMS, 2 Hz, Lee et al., 2014) as well as a Tenax cartridge sampler with subsequent GCxGC analysis for speciated BBVOCs (intermittent transect integrations, Hatch et al., 2015; Mondello et al., 2008), which we use to support mass assignments from the UW I<sup>-</sup> HR ToF CIMS for some

155 phenolic compounds (see SI).

described in (Kupc et al., 2018). The sample for the UHSAS was diluted up to a factor 2.9 for part of the flights to increase accuracy at higher concentrations. The aircraft had a standard meteorological probe (Aventech ARIM 200) for temperature, pressure, relative humidity, wind speed and direction. We use NO<sub>2</sub> photolysis rates (jNO<sub>2</sub>) collected by upward and downward facing jNO<sub>2</sub> filter radiometers (Metcon, GmbH , 1 Hz, Kupc et al., 2018; Warneke et al., 2016).

#### 2.1.2 NASA DC-8 Instrument Descriptions

- The NASA DC-8 aircraft sampled 14 wildfires in the western U.S. while based in Boise, ID as well as about 90 prescribed agricultural southeastern U.S. fires while based in Salina, KS between 22 July 2019 and 5 September 2019. Aircraft speed was 167.2 ± 3.4 m s<sup>-1</sup>, which yields a horizontal resolution of ~167 m for the in situ 1 s measurements. Similar to the NOAA Twin Otter, sampling was semi-Lagrangian. However, smoke emission times for the plume center of WF1 and WF2 covered a larger time period (~30 60 min) compared to the NOAA Twin Otter (SI Table 2).
- In this analysis we use measurements of CO from a tunable diode laser spectrometer (1 Hz, Sachse et al., 1991) when available and from a cavity enhanced spectrometer (CES, 1 Hz, Eilerman et al., 2016) when unavailable. In the fires investigated here both instruments agree well within <1 %. Measurements of NO<sub>2</sub>, NO<sub>y</sub> and O<sub>3</sub> are provided by a NOAA chemiluminescence

We use data from a commercial cavity ringdown spectrometer (Picarro G2401-m) for measurements of CO, CO<sub>2</sub>, and CH<sub>4</sub> (0.5 Hz, Crosson, 2008). We use measurements from a custom chemiluminescence instrument (1 Hz) for NO, NO<sub>2</sub> and O<sub>3</sub> (Sparks et al., 2019). Aerosol surface area measurements were collected by an ultra-high sensitivity aerosol spectrometer (UHSAS, 1 Hz, Kupc et al., 2018). The UHSAS data were corrected for coincidence up to a factor to 1.4, following the method

(CL, 1 Hz, Pollack et al., 2010; Ridley et al., 1992; Stedman et al., 1972) instrument. When measurements of  $NO_2$  by the NOAA CL instrument are unavailable we use measurements by a NOAA CES (1 Hz, Min et al., 2016). These two measurement

- methods of NO<sub>2</sub> agree within 12 % for the fires we investigate. We use measurements of NO by a laser induced fluorescence instrument (1 Hz, Rollins et al., 2020). Measurements of BBVOCs and HONO are taken from the NOAA I<sup>-</sup> ToF CIMS (1 Hz, Neuman et al., 2016; Veres et al., 2020) as well as the University of Innsbruck Proton Transfer Reaction Time of Flight Mass Spectrometer (UIBK PTR ToF MS). PAN measurements were performed by a thermal dissociation CIMS (1 Hz, Ro Lee et al., 2020). Aerosol surface area measurements are taken from a scanning mobility particle sizer and laser aerosol spectrometer
- (SMPS and LAS, 1 Hz, LAS, n.d.; Moore et al., 2021; SMPS, n.d.). Spectrally resolved actinic flux was measured with separate upward and downward facing actinic flux optics (CAFS, 1 Hz, Shetter and Müller, 1999). These fluxes were used to calculate photolysis rates using the photochemistry routine contained in the NCAR TUV model (v5.3.2).

#### 2.1.3 Plume Age Determination

Plume age estimates are made by air parcel trajectories computed in the HYSPLIT trajectory model with multiple highresolution meteorological datasets (HRRR 3 km, NAM CONUS nest 3 km, and GFS 0.25°). These estimates account for buoyant plume rise as well as horizontal advection. Uncertainties in plume age are determined from spread between the meteorological datasets, mismatch between observed and archived winds, and trajectory spatial error in missing the known fire source. Typical uncertainties are 25 % of the estimated age (Holmes et al., 2020).

#### **2.2 Fire Descriptions**

- This analysis focuses on four semi-Lagrangian experiments from three separate fire complexes: the Castle fire plume in northern Arizona, the 204 Cow fire plume in central Oregon (referred to as Cow from here on), and two from the Williams Flats fire plume in eastern Washington (referred to as WF1 and WF2 from here on). Table 1 summarizes fire locations, sampling platform, sampling times, and fuel types (Inciweb, 2019b, 2019c, 2019a). Figure 1 displays flight paths. We select the above plume samplings among others because of their data coverage, potential for active chemistry and sunset-like
- conditions defined as the following: 1) sampled by semi-Lagrangian transects roughly perpendicular to the prevailing wind direction, 2) had available measurements of CO, NO<sub>x</sub>, HONO, O<sub>3</sub>, photolysis rates, and aerosol surface area, 3) contained either reduced plume-center photolysis ( $jNO_2 < 10^{-3} s^{-1}$ ) or plume ages <3 h by sunset, and 4) sampled a plume age range >1 h. The WF fire started on 2 August 2019 and grew to a total of 179.9 km<sup>2</sup> before it was contained on 25 August 2019. The fuel was mostly short grass (~0.3 m tall) as well as ponderosa and mixed conifer timber (Inciweb, 2019c). The DC-8 aircraft
- performed three semi-Lagrangian smoke transect patterns on 7 August 2019 when the fire had burned about 101.2 km<sup>2</sup>. This study focuses on the first two sampling patterns: the WF1 (Figure 1 B) and WF2 (Figure 1 C). WF1 contained smoke emitted from about 14:00 16:00 local time (PDT), or the early to late afternoon, while the second pattern sampled smoke emitted near sunset. The sampled smoke varied in age from 36 min 4 h.

The Castle fire began on 12 July 2019 and was allowed to burn the mixed conifer fuel in a defined area that eventually reached

- 78.4 km<sup>2</sup>, and burned out on 15 October 2019 (Inciweb, 2019b). The Twin Otter aircraft performed one semi-Lagrangian transect pattern during sunset on 21 August 2019 when small pockets of remaining fuels were burning (Figure 1D). The sampled smoke varied in age from approximately 2 min 1.5 h. The Castle fire had a neighboring fire named Ikes. Smoke from the Ikes fire visually mixed (SI Figure 1) with the Castle fire plume after the fourth transect downwind of the Castle fire (Figure 1D). For that reason, this analysis focuses on the first four transects only.
- The Cow fire started on 9 August 2019 and was allowed to burn eventually reaching 39.1 km<sup>2</sup> by 15 September 2019. The fuel was mainly lodgepole pine at lower elevations and mixed conifer at higher elevations with abundant downed timber. The Twin Otter aircraft performed three semi-Lagrangian transect patterns on 28 August 2019, by which time the fire had burned 13.9 km<sup>2</sup> (Inciweb, 2019a). This study focuses on the third semi-Lagrangian transect pattern, which was conducted after sunset (Figure 1E). The sampled smoke in this analysis had aged approximately 2 3 h.

#### 215 2.3 Box Model Description

We modeled smoke plumes from three fires (Castle, Cow, and WF). We present four model cases (Castle, Cow, WF1, WF2) constrained by aircraft observations and one case (Dark) identical to the WF2 case except all modeled photolysis frequencies are set to zero. We consider the dark model run only for the WF2 case and not the others since it is a hypothetical exercise intended to illustrate the evolution of smoke emitted after dark, a case for which there were no available observations from the 2019 campaign. The Dark case is used to understand the effect of photolysis on the WF2 model run.

- 2019 campaign. The Dark case is used to understand the effect of photolysis on the WF2 model run. There were sufficient emissions for the WF1, WF2, Dark, and Cow model runs such that there were emissions remaining above background levels after 12 h of model time. The Cow, WF2, and Dark cases are run from emission until sunrise the following day (about 12 h). The Castle case is run for 2.6 h until all BB emissions are near (<<1%) background levels. We run the WF1 case until the age of the oldest sampled smoke (~4 h) because we do not have any observations of photolysis rates
- with which to constrain the model past that point.

Box modelling was performed using the Framework for 0-D Atmospheric Modelling (F0AM) (Wolfe et al., 2016) with chemistry and emissions described in the following section. We start the model at the emission time (age = 0) of the earliest smoke transect (the youngest sampled smoke), which occurred between 2 min and 2 h before the first plume transect, depending on the plume. In most cases, we use an iterative method constrained to a subset of observations (described in section 2.3.3) to

estimate emissions.

While all plumes were sampled by aircraft following a semi-Lagrangian strategy, we model each plume as if it were Lagrangian - i.e., it is assumed that the emissions and fire conditions were constant over the course of sampling. Further, we constrain our model to plume-center observations because we model only the plume-center and represent mixing through a dilution term. Consequently, the model does not represent differences in chemical regimes that may occur between the center and edge of a

235 plume. Components of our model have been used for other applications (Decker et al., 2019; McDuffie et al., 2018; Robinson et al., 2021; Wagner et al., 2013). However, the combination of the components is specific to only this work.

#### 2.3.1 Chemistry and Emissions

is not important.

Our model uses the master chemical mechanism (MCM, v3.3.1 via http://mcm.york.ac.uk), in conjunction with a NOAA biomass burning mechanism included in F0AM v4.0 (Bloss et al., 2005; Coggon et al., 2019; Decker et al., 2019; Jenkin et al.,

- 1997, 2003, 2012, 2015) and updates to OH- and NO<sub>3</sub>-initiated oxidation of phenolic compounds (Bolzacchini et al., 2001; Calvert et al., 2011; Finewax et al., 2018; Nakao et al., 2011; Olariu et al., 2002, 2013; Schwantes et al., 2017). Briefly, we update the phenol oxidation product yields of catechol, methylcatechol, and three dimethylcatechols reacting with NO<sub>3</sub> and OH. Further, we expand the phenolic oxidation pathways in the MCM from 50 to 140 reactions by extrapolating analogous branching ratios, rate coefficients and products from studies of phenol and cresol oxidation (see SI).
- We initiate the model, as discussed in section 2.3.3., using an emissions inventory of 302 BBVOCs in the form of emission ratios (ERs).

$$ER_{x} = \frac{X \text{ (ppbv)}}{CO \text{ (ppmv)}},$$
(1)

Note that an ER is used to describe an emission (when smoke age = 0) and is different than a Normalized Excess Mixing Ratio (defined in section 2.4.1) used to describe observations when smoke age > 0. The ER inventory is described in detail in Decker

- et al., 2019 and uses an average of BBVOC emission ratios of ponderosa pine fuel from the Fire Lab at Missoula Experiment (FLAME-4) (Hatch et al., 2017) and the Fire Influence on Regional and Global Environments Experiment (FIREX lab) (Koss et al., 2018) with rate coefficients taken from literature when available or estimated when unavailable. Approximately 250 BBVOCs in the inventory are not included in the MCM and do not have published mechanisms. Therefore, reactions of those compounds with NO<sub>3</sub>, OH, and O<sub>3</sub> lead to a generic product.
- The model includes heterogeneous NO<sub>3</sub> and N<sub>2</sub>O<sub>5</sub> uptake onto aerosol, calculated for NO<sub>3</sub> heterogeneous reactivity, as

$$k_{NO_3}^{aerosol} = K_{eq}[NO_2]k_{N_2O_5} + k_{NO_3+aerosol}$$
(2)

where  $k_{NO_3}^{aerosol}$  is a first order rate coefficient,  $K_{eq}$  is the equilibrium rate constant for (R4) and  $k_{NO_3+aerosol}$  is a first order rate coefficient for uptake expressed below. Note, however, that the following equation applies for small uptake coefficients and small aerosol diameters where gas phase diffusion does not limit uptake. For large particle diameters or large uptake coefficients, the simplified heterogeneous uptake equation requires a correction for gas phase diffusion (Fuchs and Sutugin, 1970; Kolb et al., 2010). For accumulation mode particles of order 100 nm and uptake coefficients of order 0.01, this correction

$$k_{x+aerosol} = \frac{\gamma \bar{c}SA}{4}$$
(3)

Here  $\gamma$  is the aerosol uptake coefficient,  $\bar{c}$  is the mean molecular speed, and SA is the measured aerosol surface area at plumecenter. We use  $\gamma_{N_2O_5} = 10^{-2}$  and  $\gamma_{NO_3} = 10^{-3}$  (McDuffie et al., 2018).

#### 2.3.2 Model Constraints

Our model is constrained to plume-center and, for some compounds, background measurements of aerosol surface area, photolysis rates,  $O_3$ , CO,  $NO_x$ , HONO, and total oxidized nitrogen ( $NO_y$ ). Measurements of  $NO_y$  are only available from the

- DC-8 measurements. We also constrain our models to the meteorological conditions pressure, temperature, and relative humidity. Fire emissions and photolysis conditions can change rapidly, therefore we constrain the model to a subset of plume transects. We chose transects that showed a monotonic decrease of CO with distance from the fire, cover an age range of at least one hour, and have similar emission times as shown in SI Figure 2 - 3 and SI Table 2.
- All model runs included a constant first-order plume dilution rate coefficient ( $k_{dil}$ ) determined by applying an exponential fit 275 to observed CO as a function of plume age (SI Figure 3). We fit only points used to constrain the model and fixed the exponential fit offset to the observed CO background. We applied  $k_{dil}$  to all species in the model. We find values of  $k_{dil}$  that range between  $1.6 - 46 \times 10^{-5}$  s<sup>-1</sup> (SI Table 3), equivalent to a lifetime ( $\tau_{dil} = 1/k_{dil}$ ) of 0.6 - 17.3 h.

Plume-center observations were determined using a "top 5 %" method as described by Peng et al., 2020. Briefly, within a transect we determine the location of the greatest 5 % of observations for CO and use that location of the plume for analysis
of other compounds. This method obtains an average observation for the center, or most concentrated, parts of the plume.

- Reported uncertainties are the 1- $\sigma$  variability of the top 5 % region and instrument uncertainties added in quadrature. Particulate matter in BB plumes attenuates sunlight, and thus photolysis rates, in a process we refer to as plume darkening. In WF plumes we use plume-center observations of 20 photolysis rates (listed in SI Table 4), but for the Castle and Cow plumes only jNO<sub>2</sub> is available due to the limited instrument payload on the NOAA Twin Otter. Average attenuation of jNO<sub>2</sub> within
- the WF1 and WF2 plumes was 96% (meaning jNO<sub>2</sub> at plume-center was 4 % of jNO<sub>2</sub> outside of the plume). Plume-center attenuation of jNO<sub>2</sub> was 29% for the Castle plume. We sample the Cow plume after sunset and therefore do not have observation of jNO<sub>2</sub> while the smoke was under sunlight (0 2 h). We estimate that plume-center jNO<sub>2</sub> attenuation was 34%. This estimate was made by comparing jNO<sub>2</sub> attenuation to plume size (by CO) in the WF and Castle model runs and is consistent with jNO<sub>2</sub> attenuation in plumes emitted from the Cow fire sampled on other days. All other photolysis rates were
- estimated using a ratio of the observed jNO<sub>2</sub> to calculated photolysis rates using an MCM trigonometric solar zenith angle (SZA) function below.

$$J = l * \cos(SZA)^m * e^{-n * \sec(SZA)}$$
(4)

Where l, m, n are derived from least squares fits to j-values from a radiative transfer model and literature cross sections/quantum yields. This calculation is a standard photolysis value method in F0AM and is described by Jenkin et al., 1997. However, this method does not account for overhead O<sub>3</sub> column, surface albedo, aerosol or other effects.

1997. However, this method does not account for overhead  $O_3$  column, surface albedo, aerosol or other effects. In all of the plumes studied here, observed jNO<sub>2</sub> rates are below  $10^{-3}$  s<sup>-1</sup> excluding the first few minutes of the Castle plume (see Figure 2). Values of jNO<sub>2</sub> in the WF2 plume remained low, near  $10^{-4}$  s<sup>-1</sup> during the sampling time. In contrast, the WF1 plume exhibits increasing jNO<sub>2</sub> rates, which eventually reach 8×10<sup>-4</sup> s<sup>-1</sup>. Differences in the photolysis rates between the first and second pass is likely due to the setting sun. Finally, observations of photolysis rates are negligible in the Cow plume as it

was sampled after sunset.

#### 2.3.3 Model Initiation

In all plumes except the Castle plume, our first transect sampled smoke 36 min - 2 h old and therefore we implemented an iterative method (McDuffie et al., 2018; Wagner et al., 2013) to estimate initial emissions (at age = 0). We began with best-

- guess estimates of CO, NO, NO<sub>2</sub>, HONO, O<sub>3</sub> and all BBVOCs (determined by CO and our emissions inventory by Eq. (1)) then systematically changed these initial conditions to minimize the differences between model output and observations downwind. Initial conditions in the Castle run were taken directly from observations of NO, NO<sub>2</sub>, O<sub>3</sub>, CO, HONO, phenol, catechol, cresol, and methylcatechol in the first transect where the plume age was  $3 \pm 1$  min, and therefore was close to age = 0. We initiated the remaining 298 BBVOCs by using CO and Eq. (1). Initial conditions for all cases are shown in SI Table
- 5Error! Reference source not found.. In all cases, backgrounds of NO, NO<sub>2</sub>, O<sub>3</sub>, CO and HONO were taken as an average outside of the plume and BBVOC backgrounds were assumed to be zero. Background mixing ratios used in all cases are shown in SI Table 3Error! Reference source not found..

We determined best-guess estimates of CO and HONO directly from observations of the first transect. To determine a bestguess estimate for NO<sub>x</sub> we used the sum of observed NO and NO<sub>2</sub> for the Cow run or NO<sub>y</sub> minus HONO (as NO<sub>y</sub> will contain

HONO) for the WF runs. Best-guess estimates of O<sub>3</sub> were determined using an average of background O<sub>3</sub> observations from a flight leg upwind of the fire and outside of the plume transects, which can vary (SI Table 6Error! Reference source not found.).

We began iteration with CO and  $k_{dil}$  by increasing best-guess estimates of CO and varying  $k_{dil}$  within the fit errors until we minimized the differences between observed and modeled CO. This in-turn determines the emissions of BBVOCs by Eq. (1).

- Next, we iterated NO<sub>x</sub>, HONO and the NO/NO<sub>x</sub> ratio such that the sum of NO<sub>x</sub> and HONO did not exceed the observed NO<sub>y</sub> and the initial NO/NO<sub>x</sub> ratio remained between 0.6 - 1 (Roberts et al., 2020). Lastly, we iterated the initial and background O<sub>3</sub>. As explained in section 2.4, we were required to iterate on background O<sub>3</sub> in some model runs in order to achieve agreement between model and observations. We repeated the above process to minimize the differences between model and observations. In an attempt to avoid finding a local solution, as opposed to the "best" solution, we reversed the order of iterating O<sub>3</sub>, NO<sub>x</sub>
- and HONO when repeating the above process.

#### 2.4 Observations and Model Comparison

Accurately modeling the first order loss of CO is critical as it determines the overall plume dilution rate coefficient and initial BBVOC mixing ratios. Median differences in modeled and observed CO range from 39.7 - 307.4 ppbv with a median difference of 2.8 - 11.7 % across all model runs. Percentage and absolute differences between the model runs and observations

are detailed in SI Table 7Error! Reference source not found. and Figure 2. Median differences of NO<sub>2</sub> and HONO are 5.1 –

32.2 % and 6.6 - 53.3 %, respectively. There are greater percentage differences in NO<sub>2</sub> and HONO that arise due to lower mixing ratio observations mostly in the WF1 and Castle plumes, with a range of absolute median differences of NO<sub>2</sub> and HONO between 0.4 - 2.0 ppbv and 0.3 - 3.4 ppbv, respectively.

Ozone median differences vary from 0.3 - 6.3 ppbv with a median difference of 0.8 - 27.2 % across all runs. For the WF1 and

- WF2 plumes we found that a significant increase ( $38.5 \pm 0.4$  and  $35.3 \pm 7.5$  ppbv, SI Table 3Error! Reference source not found. and SI Table 6Error! Reference source not found.) in model background O<sub>3</sub> compared to the upwind leg was required to capture the observed plume-center O<sub>3</sub>. This is due to photochemical O<sub>3</sub> production at the plume edges, where O<sub>3</sub> was as much as a factor of ~2 greater than the background O<sub>3</sub>. The increased plume edge O<sub>3</sub> is not captured in our plume-center model, and thus requires an increase in model background O<sub>3</sub>.
- Additional model and observation comparisons of BBVOCs, including phenolics (discussed in detail below) are included in SI Figure 5 – SI Figure 12. In most cases, the comparisons show that the model and observations agree within a factor of  $\sim$ 2, if not within observation errors.

#### 2.4.1. Comparisons of Constrained Compounds

The WF fire emissions were significantly greater than the Castle and Cow fire emissions as is seen in the observed CO (Figure
2). Initial plume-center CO was 8.26 and 8.33 ppmv in WF1 and WF2, respectively, but 2.62 and 1.95 ppmv for Cow and Castle, respectively.

We report our observations for each species (X) relative to CO in the form of normalized excess missing ratios (NEMR) following Yokelson et al., 2013 and shown in SI Figure 4.

$$NEMR = \frac{X_{Plume} - X_{Background} (ppbv)}{CO_{Plume} - CO_{Background} (ppmv)}$$
(5)

Ozone depression and negative NEMRs at the plume-center were observed in all of the sunset, nighttime or darkened fire plumes analyzed here. Observations of  $\Delta O_3/\Delta CO$  (where  $\Delta$  indicates background-corrected) in the Castle plume remains at just below background levels of  $O_3$  in all observations likely due to the small plume size and large  $O_3$  background (82.5± 2.1 ppbv). Generally,  $\Delta O_3/\Delta CO$  increases with plume age due to photochemical  $O_3$  production and mixing with background  $O_3$ . Ozone in the midday WF1 plume reaches 44.8 ± 3.4 ppbv ppmv<sup>-1</sup> of CO, or 67.4 ppbv above background,

after 
$$3.8 \pm 0.5$$
 h of transport.

Referring to SI Figure 4 we find that observed  $\Delta NO/\Delta CO$ ,  $\Delta NO_2/\Delta CO$  and  $\Delta HONO/\Delta CO$  have variable trends in all plumes. Observations of  $\Delta NO/\Delta CO$  are near zero ( $\leq 0.1$  ppbv ppmv<sup>-1</sup>) in the Castle and WF1 plumes and elevated in the WF2 and Cow plume ( $0.21 \pm 0.02 - 1.21 \pm 0.13$  ppbv ppmv<sup>-1</sup>). Observed  $\Delta NO/\Delta CO$  in the WF2 plume change sharply between the first four and last five transects suggesting changes in fire emissions or photolysis near emission. In order to avoid these changes, we

use only observations from the latter to constrain our model, as discussed in section 2.3.2.

There is a general decrease of  $\Delta NO_2/\Delta CO$  and  $\Delta HONO/\Delta CO$  over four hours of aging. Observations of  $\Delta NO_2/\Delta CO$  in the WF1 plume decrease at a faster rate than those in the WF2 plume, however, both plumes exhibit about 8.6 ppbv ppmv<sup>-1</sup> in the youngest smoke (35 ± 8 min old).

#### 2.4.2. Comparisons of P(NO<sub>3</sub>)

Emissions of NO<sub>x</sub> from biomass burning plumes provide a source of NO<sub>3</sub> suggested to be a major oxidant for BBVOCs (Kodros et al., 2020). The instantaneous NO<sub>3</sub> production rate, P(NO<sub>3</sub>), is a common metric of the potential for NO<sub>3</sub> chemistry (Brown and Stutz, 2012).

$$P(NO_3) = k_{NO_3}[NO_2][O_3]$$
(6)

At the center of the plumes presented in this study, NO<sub>3</sub> production rates were between 0.1 and 1.5 ppbv h<sup>-1</sup> as seen in Figure

- 2. These NO<sub>3</sub> production rates are consistent with those found in a nighttime agricultural smoke plume measured above a rural area at the border of Missouri and Tennessee during the South East Nexus campaign (SENEX), which varied between 0.2 and 1.2 ppbv h<sup>-1</sup> (Decker et al., 2019). These values of P(NO<sub>3</sub>) are also similar to those found in urban plumes and forested areas. Production rates of NO<sub>3</sub> in urban plumes typically range within 0 3 ppbv h<sup>-1</sup> at night but can be larger. In forested regions, P(NO<sub>3</sub>) is typically below 1 ppbv h<sup>-1</sup> at night (Brown and Stutz, 2012).
- Agreement between the model P(NO<sub>3</sub>) and observed P(NO<sub>3</sub>) reflects agreement between observed and modeled NO<sub>2</sub> and O<sub>3</sub>. The WF1 model run slightly overpredicts NO<sub>2</sub> after 3 hours of aging and therefore overpredicts P(NO<sub>3</sub>). Similarly, the Cow model run slightly underpredicts NO<sub>2</sub> compared to observations and therefore the trend in P(NO<sub>3</sub>) is slightly underpredicted.

#### 2.4.3 Comparison of phenolics

Our work focuses on the role of phenolics in BB plumes and includes updated and expanded phenolic oxidation mechanisms as described in the "Expansion of Phenolic Mechanism Description" in the SI. Therefore, capturing the phenolic evolution in our models is critical to understanding the importance of phenolics in BB. In the Castle case, which is initiated with observations of phenolics, we find excellent agreement for catechol, methycatechol, phenol, and cresol (SI Figure 5 and SI Figure 9). Further, we find that the model run lies on the upper edges of nitrocatechol errors, and the lower edge of nitrophenol errors. The model run underpredicts nitrocresol by a factor of 60. Note that we do not have available calibrations for

nitromethylcatechol, but do provide observations in arbitrary units for the purpose of comparing the time evolution of this compound.

Overall the model recreates the relative time evolution of nitrophenolics well. Disagreement between the model and observed compounds could be caused by many factors including, but not limited to, interfering isomers measured by the UW I<sup>-</sup> HR ToF CIMS or the NOAA I<sup>-</sup> ToF CIMS, variable fire ERs, and loss or production of nitrophenolics not captured by our mechanism.

The MCM includes several gas-phase loss processes of nitrophenolics, but no gas to particle partitioning. Nitrophenolics readily partition to the aerosol phase (Finewax et al., 2018). Further, the MCM does not include photolytic loss of

nitrophenolics, despite some evidence to the contrary (Sangwan and Zhu, 2016, 2018). Omitting the aerosol loss pathway may be the cause for these discrepancies. However, precisely how these differences affect the model and observation comparison is uncertain. Therefore, when analyzing gas phase nitrophenolic evolution we only consider integrated formation, as discussed

in section 3.3.2.

All other model runs were not initiated to observations of phenolics due to the older age of smoke during the first transect. Even so, in the Cow model run (SI Figure 6 and SI Figure 10) we find agreement with catechol and methylcatechol within observation errors. Modeled phenol is about a factor of 3 ( $\Delta 1.4 - 2.0$  ppbv) greater than the observations. Modeled cresol is about a factor of 10 greater than observations, while its oxidation product, nitrocresol, is 7 times less than the observations.

Models are thus able to reproduce some, but not all, phenolic observations in the Cow plume. Observations of phenolics in the WF plumes are limited to uncalibrated catechol and nitrocatechol observations from the NOAA I<sup>-</sup> ToF CIMS (SI Figure 7 – SI Figure 8 and SI Figure 11 – SI Figure 12). In the WF1 model run, catechol and nitrocatechol appear to deplete faster than the model would suggest. The time evolution of nitrocatechol in the WF2 plume agrees well with the model, and in the WF1 model run the model matches the rough timing of the observed maximum signal.

#### 405 **3 Results and Discussion**

#### 3.1 Reactivity

Instantaneous reactivity, Eq. (7) referred to simply as reactivity here on, is used as a simplified metric to predict the competition of reactions between oxidant and BBVOC

$$k_X = \sum_i k_{X+BBVOC_i} [BBVOC_i] \tag{7}$$

where,  $k_{X+BBVOC}$  is a bimolecular rate coefficient for the reaction of X + BBVOC (where X is O<sub>3</sub>, NO<sub>3</sub> or OH) and  $k_X$  is an instantaneous first order rate coefficient. Here, we calculate and detail the reactivity for O<sub>3</sub>, NO<sub>3</sub> and OH oxidation of BBVOCs to understand their predicted competition. We also discuss how reactivity of the BB plumes studied here compare to other environments.

At emission, BBVOCs account for the majority of total reactivity for OH (87.7 %), NO<sub>3</sub> (80.1 %), and O<sub>3</sub> (99.6 %) as seen by the bars in Figure 3. HCHO and CO account for 5.1 % and 5.3 % of OH reactivity, respectively while NO<sub>2</sub> accounts for a small (0.3 %) fraction. In this analysis we do not specify an aldehyde group, and therefore separate HCHO from the general BBVOC groupings. We exclude O<sub>3</sub> reactivity to NO in Figure 3 because during the daytime this reaction is in a rapid cycle with NO<sub>2</sub> photolysis and regeneration of O<sub>3</sub> in which odd oxygen,  $O_x = NO_2 + O_3$ , is conserved. Further reactions of O<sub>3</sub> and NO<sub>2</sub> can lead to loss of O<sub>x</sub>. This analysis includes BBVOC oxidation by O<sub>3</sub> but not a detailed budget for O<sub>x</sub>.

Underneath each reactivity bar in Figure 3 we show the partitioning of the initial BBVOC reactivity. Almost three quarters of OH reactivity is from alkenes (33.0 %), furans (25.0 %) and phenolics (16.4 %). The reactivity of NO<sub>3</sub>, by contrast, is

controlled by phenolics (64.4 %) and O<sub>3</sub> reactivity is controlled by alkenes (53.8 %) and terpenes (39.2 %). Nitrate radical reactivity toward a smaller fraction of VOCs is consistent with other reactivity analyses of OH, NO<sub>3</sub> and O<sub>3</sub> in forest air (Palm et al., 2017).

Below each pie chart in Figure 3 we show reactivity for OH, NO<sub>3</sub>, and O<sub>3</sub> toward BBVOCs on an absolute scale. As BBVOCs are oxidized and the plume dilutes the plume reactivity is reduced. Decay of OH and NO<sub>3</sub> reactivity is nearly identical, while that of O<sub>3</sub> is different (e.g., WF2 and Dark). As a result, fewer BBVOCs, specifically alkenes, are oxidized in the Dark model run keeping reactivity greater when compared to the WF2 model run.

Total initial OH reactivity toward BBVOCs ranges from  $98.3 - 450.0 \text{ s}^{-1}$ . Since the modeled total reactivity is proportional to 430 the plume's initial emission of CO, the largest plumes, WF and Dark, have the greatest total initial total reactivity. Typical OH reactivities range between  $7 - 130 \text{ s}^{-1}$  for urban plumes or  $1 - 70 \text{ s}^{-1}$  in forests (Yang et al., 2016), demonstrating that wildfire plumes can be similar to urban plumes or significantly more reactive.

Total initial O<sub>3</sub> reactivity toward BBVOCs ranges between  $1 \times 10^{-4}$  s<sup>-1</sup> and  $6 \times 10^{-4}$  s<sup>-1</sup>. A recent study of a suburban site in China found O<sub>3</sub> reactivities toward non-methane VOCs between  $2.5 \times 10^{-7} - 1.1 \times 10^{-6}$  s<sup>-1</sup> (Yang et al., 2020). Reactivity in wildfire plumes exceeds that in urban plumes by a factor of 80 – 3000.

Total initial NO<sub>3</sub> reactivity toward BBVOCs ranges from 17.1 – 70.3 s<sup>-1</sup>. Reactivity of NO<sub>3</sub> is typically reported as a lifetime  $(\tau_{NO_3})$ , which is the NO<sub>3</sub> concentration over the NO<sub>3</sub> production rate under the assumption of a steady state in both NO<sub>3</sub> and N<sub>2</sub>O<sub>5</sub> (Brown et al., 2003). Since NO<sub>3</sub> and N<sub>2</sub>O<sub>5</sub> readily interconvert (R4), the sum of  $\tau_{NO_3}$  and  $\tau_{N_2O_5}$  are reported.

$$\tau_{NO_3+N_2O_5} = \frac{NO_3+N_2O_5}{P(NO_3)}$$
 (8)

Using Eq. (8), modeled steady-state lifetimes are predicted to be between 0.5 - 1.2 s. Typical  $\tau_{NO_3}$  in urban plumes range from tens of seconds to tens of minutes and  $\tau_{NO_3}$  in forested regions have been reported between 20 s – 15 min (Brown and Stutz, 2012). The reactivity of NO<sub>3</sub> in wildfire plumes sampled during FIREX-AQ is  $10 - 10^4$  times greater than typical values in forested or urban environments. The increased reactivity of NO<sub>3</sub> to BBVOCs within wildfire plumes is greater than the increased reactivity for OH and O<sub>3</sub>, highlighting that BB plumes have large overall reactivity that is more pronounced for NO<sub>3</sub> than other oxidants. The increased reactivity of NO<sub>3</sub> is due to the specific emissions from biomass burning, such as phenolics and furans that have substantial reactivity towards NO<sub>3</sub>. The compounds greatly increase NO<sub>3</sub> reactivity compared to urban VOC profiles, but do not increase OH reactivity to the same degree.

In addition to a large suite of reactive BBVOCs that increase NO<sub>3</sub> reactivity, smoke contains concentrations of aerosol and 450 aerosol surface area that are far greater than normally found in urban areas (Decker et al., 2019). When considering NO<sub>3</sub> reactivity we must also consider aerosols, since aerosols present a loss pathway for NO<sub>3</sub> and its equilibrium product  $N_2O_5$  (Brown and Stutz, 2012; Goldberger et al., 2019; Tereszchuk et al., 2011). As explained in section 2.3.1 we calculate the  $NO_3$  heterogeneous reactivity to understand the competition between  $NO_3$  loss to BBVOCs and  $NO_3/N_2O_5$  heterogeneous loss to reaction with aerosol.

As shown in SI Figure 13 heterogeneous losses of NO<sub>3</sub> and N<sub>2</sub>O<sub>5</sub> are  $<\sim$ 2.5 % of total NO<sub>3</sub> reactivity in all model runs. Further, we find that >90 % of aerosol loss is through N<sub>2</sub>O<sub>5</sub> rather than NO<sub>3</sub> uptake. Therefore heterogeneous losses of NO<sub>3</sub> and N<sub>2</sub>O<sub>5</sub> do not appreciably compete with gas phase BBVOC oxidation, consistent with a similar analysis of nighttime smoke plumes (Decker et al., 2019).

While our analysis finds that the reactivity in a BB plume is far greater than other environments, it is important to note that our calculations use a large suite of the most reactive VOCs that may not be included in other reactivity studies. Further, our reactivity calculations are based on our BBVOC ER and kinetic database as described by Decker et al., 2019. While this database includes rate coefficients for the most reactive BBVOCs, it does not include rate coefficients for all 302 BBVOCs with all oxidants. Therefore, our reactivity estimates may be a lower estimate. Our VOC profile does not include alkanes, since FIREX lab studies (Hatch et al., 2015; Koss et al., 2018) and an OH reactivity analysis of FIREX lab emissions found that OH

reactivity toward alkanes accounted for 0 - 1 % of total BBVOC reactivity across all fuels (Gilman et al., 2015). Therefore, we expect the absent alkane reactivity in this study to be negligible.

#### **3.2 Oxidation Rates**

While reactivity is a useful metric to predict the competition between reactions, it does not account for oxidant concentration, which can vary widely depending on photolysis rates, emissions, and competing oxidants. In the following sections we discuss the BBVOC oxidation rate, which is related to reactivity through the oxidant concentration as shown below

 $R_{X} = \sum k_{X+BBVOC_{i}} [BBVOC_{i}][X] = k_{X}[X]$ (9)

where  $R_X$  is the BBVOC oxidation rate,  $k_X$  is the biomolecular rate coefficient between X and BBVOC, and X is OH, NO<sub>3</sub> or O<sub>3</sub>. In the following sections we compare and contrast reactivity and oxidation budgets and discuss how the initial reactivity changes with plume age for different BBVOC groups. Finally, we discuss the oxidant competition between NO<sub>3</sub>, OH, and O<sub>3</sub> for three main groups of BBVOCs: phenolics, furans/furfurals, and alkenes/terpenes.

**3.2.1 Oxidation of BBVOCs** 

# The integrated oxidation rate, or the oxidation budget (Figure 4), is similar to initial reactivity shown in Figure 3 for OH oxidation suggesting initial reactivity may be a good indicator for integrated reactivity. However, this does not hold true for $NO_3$ or $O_3$ .

The initial NO<sub>3</sub> reactivity differs substantially from the oxidation budget. For example, 20 % of initial NO<sub>3</sub> reactivity is due to NO, but NO accounts for  $\leq 1$  % of integrated NO<sub>3</sub> loss. Further, photolysis of NO<sub>3</sub> accounts for <1 % of NO<sub>3</sub> loss in all model runs and is greatest in the Castle plume (0.6 %) where measured jNO<sub>2</sub> and calculated jNO<sub>3</sub> reached maximum values of  $4 \times 10^{-1}$   $^{3}$  and 0.14 s<sup>-1</sup>, respectively. Although daytime NO<sub>3</sub> oxidation of reactive VOCs has been found for heavily polluted urban air (Brown et al., 2005; Geyer et al., 2003; Osthoff et al., 2006), the dominant NO<sub>3</sub> loss processes in urban plumes is NO reaction

and photolysis (Brown and Stutz, 2012; Wayne et al., 1991). The different controlling NO<sub>3</sub> loss pathway here highlights the unique and highly reactive environment of BB plumes. Further, 67 - 70 % of integrated NO<sub>3</sub> reaction is due to phenolics, which is larger than initial total NO<sub>3</sub> reactivity (56 %). Integrated alkene, terpene, and furan oxidation by NO<sub>3</sub> are all lower than their initial reactivities.

The production of NO<sub>3</sub>, by (R3), and subsequent loss to BBVOCs is a significant (8 - 21 %) loss of O<sub>3</sub>, and much greater than

- the initial O<sub>3</sub> reactivity to NO<sub>2</sub> of 0.4 %. Similarly, integrated loss of O<sub>3</sub> to alkenes (40 49 %) and terpenes (16 23 %) is much less than initial reactivity would suggest (54 % and 39 %, respectively). Conversely, phenolics and furans account for 4 - 11 % and 13 - 20 % of O<sub>3</sub> loss, respectively, even though their relative initial reactivity is < 1 % and 7 %, respectively. Overall, the differences between initial reactivity and integrated oxidation rate are explained by changing reactivity as BBVOC are oxidized with plume age.
- An example is seen in Figure 5 for O<sub>3</sub> in the Castle model run, which has a large O<sub>3</sub> background ( $72 \pm 1 82 \pm 2$  ppbv), is a relatively small plume, and is sunlit at emission. As a result, alkenes and terpenes are depleted quickly through oxidation by O<sub>3</sub> and OH. The combined O<sub>3</sub> reactivity of alkenes and terpenes reduces from 82 % to 44 % after two hours, during which time phenolic reactivity increases from < 1 % to ~40 %. In other words, as BBVOCs are depleted the reactivity profile of each oxidant will change and can result in significant differences between the initial reactivity and oxidant budget.
- In contrast to NO<sub>3</sub> and O<sub>3</sub>, loss of OH by each BBVOC group is within 1 % of that predicted by the initial reactivity, except for terpenes. Initial reactivity of terpenes is about 13 %, while actual destruction of OH by terpenes averaged to 8 %. While terpene oxidation by OH is lower than its reactivity in all model runs, it is especially low (2 %) in the WF1 model run, which is likely due to the large concentration of O<sub>3</sub> from photochemical production.
- Losses of OH are not only due to highly reactive BBVOCs. HCHO, CO, and NO<sub>2</sub> are responsible for 12 14 % of OH destruction. This is consistent with an OH reactivity analysis from North American fuels burned during the FIREX laboratory study, which found  $13 \pm 1$  % of OH reactivity was due to HCHO, CO, and NO<sub>2</sub> (Gilman et al., 2015). The fraction of OH reactivity toward CO and NO<sub>2</sub> are similar to those found in a tropical rainforest (Fuchs et al., 2017), but much smaller than the fraction of OH reactivity toward CO (7 %) and NO<sub>2</sub> (18 %) found at an urban site (Gilman et al., 2009) and the fraction of OH reactivity toward CO (20 - 25 %) and NO<sub>x</sub> (12 - 22 %) at a rural site (Edwards et al., 2013).

#### 510 3.2.2 Oxidant Competition

To study the competition between all oxidants, we focus on three main BBVOC groups: phenolics, furans/furfurals, and alkenes/terpenes. Generally, furans/furfurals and alkenes/terpenes groups are mainly oxidized by OH and O<sub>3</sub>, while NO<sub>3</sub> plays a small role (Figure 6). Oxidation of furans/furfurals and alkenes/terpenes by OH (18 - 55 %, 11 - 43 %, respectively) and O<sub>3</sub> (39 - 76 %, 54 - 88 %, respectively) can vary widely depending on the plume. We find this is due to the variability of actinic

flux. In model runs with less photolysis at emission, OH oxidation is low compared to model runs that are more optically thin.

This reduction of oxidation by OH appears to be replaced by  $O_3$  rather than  $NO_3$ . For example, relative furan/furfural oxidation by OH in the WF1 model run (relatively large integrated j $NO_2$ ) is 31 % less than that in the Cow model run (comparatively lower integrated j $NO_2$ ), yet  $O_3$  oxidation is 32 % greater.

This relationship does not hold for phenolics, which are subject to significant NO<sub>3</sub> oxidation (26 - 52 %) (Figure 6). Phenolic

oxidation by OH (22 - 43 %) and O<sub>3</sub> (16 - 33 %) are slightly less than NO<sub>3</sub>. As a result, phenolic oxidation by NO<sub>3</sub> dominates in the WF1 and Dark model runs, while OH dominates in the Castle model run. In the WF2 and Cow model runs, NO<sub>3</sub> and OH oxidation is roughly equal.

Generally,  $NO_3$  oxidation of phenolics increases with  $O_3$  availability and decreases with available actinic flux, but these relationships are coupled and complex. One example is seen in the WF2 model run, which has the second lowest integrated

jNO<sub>2</sub> value, and large emissions of NO that keep O<sub>3</sub> low during sunlit hours. Therefore, P(NO<sub>3</sub>) is reduced, NO<sub>3</sub> is present at lower mixing ratios within the first hour of oxidation, and phenolics are less subject to NO<sub>3</sub> oxidation when compared to the other model runs.

As actinic flux increases so does OH and  $O_3$  production, and therefore oxidant competition. One example is shown by the Castle model run where OH leads phenolic oxidation (41 %) with  $O_3$  second (33 %). The Castle model run demonstrates the

530 greatest observed background O<sub>3</sub> (90 ppbv). Further, the Castle model run has significantly smaller total emissions (based on CO) than the other model runs and the greatest integrated jNO<sub>2</sub>. Due to the increased background O<sub>3</sub> and photochemical production of OH, NO<sub>3</sub> plays a smaller role in the oxidation of phenolics (Akherati et al., 2020).

#### 3.3 Phenolic Oxidation and Nitrophenolic Production

The importance of phenolic oxidation for BB is evidenced by the rapidly growing literature (Bertrand et al., 2018; Chen et al., 2019; Coggon et al., 2019; Decker et al., 2019; Finewax et al., 2018; Gaston et al., 2016; Hartikainen et al., 2018; Iinuma et al., 2010; Lauraguais et al., 2014; Lin et al., 2015; Liu et al., 2019; Meng et al., 2020; Mohr et al., 2013; Palm et al., 2020; Selimovic et al., 2020; Wang and Li, 2021; Xie et al., 2017). Both OH and NO<sub>3</sub> oxidation of phenolics leads to nitrophenolics, which have been shown to significantly contribute to SOA production (Palm et al., 2020). However, not all nitrophenolics are created equal. Understanding the competition between phenolic oxidation by NO<sub>3</sub> and OH is critical because their oxidation 540 pathways have significantly different implications for nitrogen budgets and total nitrophenolic yield. Nitrophenolics formed by OH requires one NO<sub>2</sub> molecule with a nitrophenolic yield between 27 – 33 %. In contrast nitrophenolics formed by NO<sub>3</sub>

require two molecules of NO<sub>2</sub>, have a yield of 85 - 97 % and produce HNO<sub>3</sub> as a byproduct (see SI Figure 14 and Finewax et al., 2018).

Yet, current phenolic mechanisms are extremely limited. For example, in the MCM nitrophenolics are the only oxidation products of phenolics + NO<sub>3</sub> or OH and the yields are assumed to be 100%. Phenolic oxidation studies are typically limited to final products without detailed examination of intermediates. Phenol and cresol reactions are well studied in comparison to catechol, methylcatechol, and higher order phenolics. For that reason, we use studies of phenol and cresol oxidation to extrapolate analogous branching ratios, rate coefficients, and products for catechol, methylcatechol, and three isomers of dimethylcatechol. All of these compounds are included in the MCM, but for the purpose of the following analysis we have expanded the phenolic reaction pathways in our model as explained in the SI and shown in SI Figure 14.

In the remaining sections, we detail how the competition for phenolic oxidation changes as the plume evolves over time. We then discuss the factors that cause differences in nitrophenolic production rate as well as how differences in OH and NO<sub>3</sub> phenolic oxidation lead to substantial differences in nitrocatechol yield. Finally, in the following section, we explore how nitrophenolics significantly impact the nitrogen budget.

#### 555 3.3.1 Evolution of Phenolic Oxidation

Generally, the modeled total phenolic oxidation rate varies between 1-10 ppbv hr<sup>-1</sup> at emission (Figure 7 A – D), but the change in oxidation rate is not constant and trends with available actinic flux. Model runs with active initial photochemistry (Castle, WF2, and Cow) exhibit decreasing total oxidation rates, while model runs with little to no photolysis (WF1 and Dark) reach a local maximum rate after ~2 h and ~5 h, respectively. These increases in oxidation rate are due to increases in O<sub>3</sub> and NO<sub>3</sub>

oxidation once NO is depleted. Generally, the phenolic lifetime increases with decreasing actinic flux. The contrast between day and night phenolic oxidation is best seen by comparing the WF2 and Dark model runs. Phenolic lifetimes in the Dark model run are, on average, a factor of ~2 greater than phenolic lifetime in the WF2 model run.

Before sunset and in early stages of plume oxidation, the major channel of phenolic oxidation is via OH. However, in the WF1 model run NO<sub>3</sub> oxidation dominates after only 12 minutes (Figure 7 A). As the WF1 model run dilutes, photolysis rates increase

- and O<sub>3</sub> is entrained promoting O<sub>3</sub> and NO<sub>3</sub> production. This increase in oxidant concentration keeps phenolic oxidation > 1 ppbv h<sup>-1</sup> for at least four hours before the end of the model (see section 2.3), unlike other model runs that drop below 1 ppbv h<sup>-1</sup> of total phenolic oxidation within 0.5 - 3 h. After 2.6 h, in the WF1 model run, all oxidants contribute equally to phenolic oxidation and thereafter, OH and O<sub>3</sub> equally split oxidation while the influence of NO<sub>3</sub> decreases. At the end of the WF1 model run, 69% of initial phenolics remain unoxidized (SI Figure 15).
- As the sun sets in our sunset model runs (WF2, Castle, and Cow) a transition from OH controlled to a mixture of NO<sub>3</sub> and O<sub>3</sub> controlled oxidation occurs when OH production, and total oxidation rate decrease rapidly. Interestingly, OH dominates phenolic oxidation in the Dark model run (initiated after sunset) for the first 1.8 h before NO<sub>3</sub> oxidation takes over. During this time, OH is produced by decomposition of Criegee intermediates formed through ozonolysis of unsaturated hydrocarbons, primarily catechol (SI Figure 14), methylcatechol and limonene. In other sunset model runs, OH plays a smaller role after
- sunset. Even so, this suggests that all BBVOC oxidation after sunset is driven by O<sub>3</sub> chemistry, either through direct oxidation by O<sub>3</sub>, NO<sub>2</sub> + O<sub>3</sub> to form NO<sub>3</sub>, or by formation and decomposition of Criegee intermediates to form OH. The WF2, Dark, and Cow model runs all contain unreacted phenolic emissions at sunrise the following day (48%, 61%, and 8%, respectively, SI Figure 15). The WF2 and Dark model runs have significantly more phenolics that remain at sunrise because of their larger (~×3) emissions compared to the Cow model run. Further, the WF2 and Dark model run conditions
- differ only by the presence of photolysis and therefore the difference in remaining phenolics between the WF2 and Dark is

due to the time of day the smoke was emitted. In contrast to these three model runs, the emissions in Castle are depleted within 2.6 h due to its small size.

#### 3.3.2 Total Nitrophenolic Formation

- Nitrophenolic formation increases with  $O_3$  and photolysis, which promotes formation of NO<sub>3</sub> and OH. For example, the Castle and Cow model runs have relatively large  $O_3$  and jNO<sub>2</sub> at emission and therefore form nitrophenolics rapidly (0.6 – 1.4 ppbv h<sup>-1</sup> within the first 15 min). In contrast, the WF and Dark model runs have near zero  $O_3$  due to large emissions of NO and relatively low or zero jNO<sub>2</sub> and therefore form nitrophenolics more slowly (<0.1 – 0.7 ppbv h<sup>-1</sup> within the first 15 min). Despite the rapid formation of nitrophenolics in the Castle model run, it has the least (excluding WF1) total nitrophenolic formation relative to total emissions as seen in Figure 8. Figure 8 shows integrated nitrophenolic formation per emitted ppmv
- of CO, which allows us to compare total nitrophenolic formation across varying plume sizes. In contrast to the Castle model run, the Cow model run has the greatest nitrophenolic formation. These differences are the result of differing phenolic oxidation pathways. The Castle model run has a large (90 ppbv) O<sub>3</sub> background, which results in O<sub>3</sub> accounting for ~40% of phenolic oxidation between 30 min 2 h of age (Figure 7 C). At the end of the Castle model run (2.6 h) O<sub>3</sub> oxidation accounts for 33% of total phenolic loss, the largest of any model run (Figure 6). This is markedly different than the Cow model run where OH and NO<sub>3</sub> chemistry control phenolic oxidation before sunset, and NO<sub>3</sub> after. While O<sub>3</sub> accounts for only 16% of phenolic loss
- at the end of the model run (~12 h). In our model, the reaction of  $O_3$  + phenolics forms a ring opening product (SI Figure 14), but the rate coefficients and mechanisms are largely uncertain as discussed in the following section.

We include 157 phenolics in our above analysis, but only a few phenolics account for large fractions of nitrophenolic formation. At the end of our model runs, catechol and methylguaiacol account for the largest fraction of phenolic oxidation. Both 600 compounds are mostly oxidized by NO<sub>3</sub>. Catechol + NO<sub>3</sub> alone accounts for 10 - 16 % of total phenolic oxidation rate or 30

-32 % of NO<sub>3</sub> + phenolic oxidation. Similarly, methylguaiacol accounts for 22 – 26 % of NO<sub>3</sub> + phenolic rates and is the largest fraction of phenolic oxidation by OH (17 – 18 % of OH + phenolic rates). However, to our knowledge, oxidation products of methylguaiacol by OH and NO<sub>3</sub> are unknown, but likely lead to nitrophenolics and therefore our nitrophenolic formation rates are likely underestimated.

#### 605 **3.3.3 Nitrocatechol Yield**

The reaction of OH and NO<sub>3</sub> with catechol to form nitrocatechol accounts for the largest fraction (32 - 33 %) of total nitrophenolic formation. Therefore, here, we focus on nitrocatechol and detail the nitrocatechol yield from NO<sub>3</sub> and OH + catechol. Understanding nitrocatechol yield and its sensitivities is important to understanding the fate of NO<sub>x</sub> and NO<sub>x</sub> lifetime discussed in the final sections. However, the nitrocatechol yield depends on many variables such as the concentrations of NO<sub>x</sub>,

BBVOC, O<sub>3</sub> and the NO<sub>x</sub>/BBVOC ratio as well as the certainty in our chemical mechanisms. Therefore, we discuss the sensitivity of all of these factors on nitrocatechol yield below.

Yields of nitrocatechol vary between 33 - 45 % depending on the model run, where NO<sub>3</sub> is responsible for 72 - 92 % of nitrocatechol (Figure 9 A). Figure 9 explores factors that govern nitrocatechol yield, defined as the molar ratio of nitrocatechol production to catechol destruction. Yields of nitrocatechol from OH are low relative to NO<sub>3</sub> yield due to the formation of

615 trihydroxybenzene and benzoquinones (SI Figure 14), which account for 10 - 32 % and 4 - 5 % of total catechol loss, respectively.

The largest yield (45 %) is from the Dark model run, where NO<sub>3</sub> oxidation accounts for more than 52 % of phenolic oxidation. In contrast, the lowest yield of nitrocatechol is from the Castle model run (33 %), which has the lowest emissions of NO<sub>x</sub> compared to the other model runs. A similar yield (34 %) is found in the WF1 model run, however this model ends after only

620 4 h when 69 % of phenolics still remain. In short, nitrocatechol yield increases with increasing fraction of phenolic oxidation by NO<sub>3</sub>.

To understand the dependence of nitrocatechol formation on  $O_3$ ,  $NO_x$ , total BBVOC emissions (defined by the sum of ERs in our BBVOC inventory) and BBVOC/NO<sub>x</sub>, we ran a sensitivity analysis on the nitrocatechol yield (Figure 9 B – E). Based on emitted NO<sub>x</sub> and CO, BBVOC/NO<sub>x</sub> ratios in plumes we sample range from 11 – 35. However, due to fire variability, BBVOC

emissions can vary by at least a factor of two and for many BBVOCs by more than a factor of 10 from our emission ratios (Decker et al., 2019). Furthermore, we only account for BBVOCs that are most reactive to O<sub>3</sub>, OH, and NO<sub>3</sub>, which is smaller than total emitted BBVOCs.

The nitrocatechol yield generally decreases with increasing  $BBVOC/NO_x$  (color scale and white lines in Figure 9 B). As expected, nitrocatechol yields increase with increasing  $NO_x$  (Figure 9 C). Across all model runs, the nitrocatechol yield

- increases to 43 % 57 % over a NO<sub>x</sub> range of 4.2 91.2 ppbv. Further, the nitrocatechol yield changes to 27 % 50 % (Figure 9 D) when varying total BBVOC emissions by a factor from 4 to 0.5. Finally, we investigate the sensitivity of nitrocatechol yield to initial O<sub>3</sub> and find that all model runs have little sensitivity to O<sub>3</sub> (Figure 9 E) with an absolute change in nitrocatechol yield <3 % for all model runs when varying initial O<sub>3</sub> over a range of 0 113 ppbv.
- The low sensitivity of nitrocatechol yield to  $O_3$  may be partially explained by competition between  $O_3$  and  $NO_3$  + phenolic reactions after sunset. To explore this, we use framework developed by Edwards et al., 2017. Briefly, as stated in section 3.2.1, BBVOCs are the main sink for NO<sub>3</sub> and therefore NO<sub>3</sub> loss rate is controlled by the NO<sub>3</sub> formation rate. As a result, NO<sub>3</sub> can be considered to be in approximate steady state between production by NO<sub>2</sub> + O<sub>3</sub> and loss by NO<sub>3</sub> + BBVOC. Further, according to Figure 4, the majority of NO<sub>3</sub> is lost to phenolics. As a result, the rate of phenolic oxidation after sunset (when OH oxidation of phenolics is minimized) can be approximated as
- $640 \quad -\frac{d[\text{phenolics}]}{dt} \approx \left(k_{0_3}[\text{phenolics}] + k_{N0_2+0_3}[N0_2]\right)[0_3] \tag{9}$

which shows that the dominant oxidant is determined by the ratio of NO<sub>2</sub> and phenolics. We find that the ratio of phenolics to NO<sub>2</sub> at which NO<sub>3</sub> and O<sub>3</sub> oxidation is equal to be ~10 (at 298 K, using an ER weighted average  $k_{O_3} = 2.6 \times 10^{-18}$  cm<sup>3</sup> molecule<sup>-1</sup> s<sup>-1</sup>) with NO<sub>3</sub> oxidation more important below this ratio, and O<sub>3</sub> oxidation more important above it. Modeled phenolics/NO<sub>2</sub>

ratios at sunset range between 0.7 - 1.2 and in all model runs, except the Castle model run, the ratio decreases with age. This suggests that in all model runs NO<sub>3</sub> oxidation is expected to control phenolic oxidation after sunset.

- The phenolic oxidation analysis above relies on phenolic mechanisms and rate coefficients that are highly uncertain. For example, the above calculated ratio could be much lower in cold lofted plumes, but knowledge of temperature dependent  $O_3$ + phenolic rate coefficients ( $k_{O_3}$ ) are unavailable. Using temperatures observed in the WF2 plume (~268 K) for  $k_{NO_2+O_3}$  (but using  $k_{O_3}$  at 298 K) the phenolics to NO<sub>2</sub> ratio at which NO<sub>3</sub> and O<sub>3</sub> oxidation is equal would be ~ 4.
- The rate coefficient and products for the reaction of catechol +  $O_3$  that we use are generated using MCM mechanism methodology (Jenkin et al., 2003; Saunders et al., 2003). An experimental study on the gas-phase reaction of catechol +  $O_3$ finds an RH dependent rate coefficient that decreases non-linearly from  $1.3 \times 10^{-17}$  to  $1.2 \times 10^{-19}$  cm<sup>3</sup> molecule<sup>-1</sup> s<sup>-1</sup> with increasing RH (El Zein et al., 2015). The MCM uses a rate coefficient of  $9.2 \times 10^{-18}$  cm<sup>3</sup> molecule<sup>-1</sup> s<sup>-1</sup>. Further, to our knowledge there are no experimental kinetic or mechanistic studies of phenol +  $O_3$ . In the plumes we investigate, RH varied
- between roughly 20 60 %. Using an RH dependent rate coefficient for O<sub>3</sub> + catechol we find that the nitrocatechol yields range between 31 - 58 % with little change in yield for the Castle model run (-2 %) and larger change for the Dark model run (+13 %).

#### 3.4 Fate of NO<sub>x</sub> in Dark BB Plumes

Fire emissions are concentrated sources of NO<sub>x</sub>, but as a result of photochemistry and oxidation the loss processes and lifetime

of plume NO<sub>x</sub> are variable. Photochemical NO<sub>x</sub> loss pathways include reaction with OH (R8), net formation of peroxy acyl nitrates (PANs) (R9), and formation of organic nitrates (R10).

$$NO_2 + OH + M \rightarrow HNO_3 + M$$
 (R8)

$$NO_2 + R(O)O_2 + M \rightarrow PAN + M \tag{R9}$$

$$NO + RO_2 + M \rightarrow RONO_2 + M \tag{R10}$$

- The NO<sub>x</sub> rate consumption is further influenced by the formation and the subsequent fate of NO<sub>3</sub> (R1 4, 6 7). Heterogeneous uptake of N<sub>2</sub>O<sub>5</sub> (R5) and production of nitrophenolics double the NO<sub>x</sub> consumption rate since in both cases subsequent chemistry consumes one additional NO<sub>2</sub> molecule, with the rate limiting step being (R3). Below, we focus on the products of NO<sub>x</sub> oxidation, determined as NO<sub>z</sub> = NO<sub>y</sub> – NO<sub>x</sub>.
- Results are similar for all model runs, and we discuss the WF2 model run as a case study. While a complete NO<sub>z</sub> budget analysis constrained to observations is beyond the scope of this work, we compare our model results of PAN, (peroxy acetyl nitrate, a component of PANs) to observations (SI Figure 8 and SI Figure 12). PAN accounts for ~65% of PANs, and PANs account for the largest fraction of NO<sub>z</sub> in our model runs during sunlit hours. Our model reproduces PAN well in one transect, but underpredicts PAN by a factor of ~2.5 in others. Similar to O<sub>3</sub> (section 2.3.2), PAN is enhanced on plume edges and the enhancement likely mixes into the center, which is not captured by our model runs. Therefore, we constrain our model to PAN

observations, present an average result (Figure 10), and consider our model unconstrained to PAN to be a lower-bound PAN estimate and our model constrained to PAN to be an upper-bound PAN estimate.

#### 3.4.1 NOz Budgets

The late day emitted plumes modeled in this paper exhibit photochemical loss of  $NO_x$  initially. In the period prior to sunset, PANs and PNA (peroxynitric acid,  $HO_2NO_2$ ) dominate  $NO_z$  and PANs alone accounts for  $51 \pm 6$  % of  $NO_z$  by sunset. The

- WF2 plume is lofted, and therefore cold (~267 K), which results in a long PAN and PNA lifetime (~150 h, and ~0.4 h, respectively, calculated from the model directly (Atkinson et al., 2006)). Even so, as these plumes continue to age, PANs and PNA decompose slowly (Figure 10) to provide NO<sub>2</sub> that promotes nitrophenolic formation and increases nitrophenolic yield (see section 3.3.3). The increase in NO<sub>2</sub> after sunset promotes methyl peroxy nitrate (CH<sub>3</sub>O<sub>2</sub>NO<sub>2</sub>) as well as NO<sub>3</sub> chemistry products, which grow steadily overnight. The contribution of PANs and PNA to NO<sub>z</sub> decreases from 71 ± 6 % at sunset to 17
- ± 2 % at sunrise. Relative NO<sub>x</sub> loss to PANs and PNA is mostly replaced by the formation of nitrophenolics (Δ 19 ± 1 %), HNO<sub>3</sub> by NO<sub>3</sub> chemistry (Δ 22 %), and other or unknown NO<sub>3</sub> products (Δ 11 %) overnight. After sunset NO<sub>3</sub> chemistry takes over and by sunrise NO<sub>3</sub> chemistry products lead the (66 ± 2 %) NO<sub>z</sub> budget. Nitrophenolic formation accounts for 56 ± 2 % of NO<sub>z</sub> in the form of HNO<sub>3</sub> and nitrophenolics where nitrophenolics alone account for 29 + 1 % of NO<sub>z</sub>. Total HNO<sub>3</sub> formation accounts for 31 + 1 % of NO<sub>z</sub>, however most (88 %) of HNO<sub>3</sub> results from NO<sub>3</sub>
- chemistry. Despite accounting for only 9% (by mole) of initial emissions in our model runs, phenolics have a large and disproportionate effect on NO<sub>x</sub> loss at night.

A similar example is seen in the Dark model run (SI Figure 18), where PANs and PNA dominate  $NO_z$  budget for 2.3 h until NO is depleted. At this time, PNA and PANs steadily decrease while  $NO_3$  products steadily increase throughout the night. By sunrise the next day,  $NO_3$  chemistry products (including unknown products) account for 80 % of  $NO_z$ . In all model runs there

is a significant (12 - 16 %) NO<sub>z</sub> formed through NO<sub>3</sub> chemistry that leads to unknown products. These unknown products are primarily the result of NO<sub>3</sub> + heterocycles such as furans and pyrroles, which have published rate coefficients but little mechanistic work in the literature.

Our  $NO_z$  budget generally agrees with the  $NO_y$  budget of western U.S. wildfire smoke sampled during the 2018 Western Wildfire Experiment for Cloud Chemistry, Aerosol Absorption, and Nitrogen (WE-CAN) presented by Juncosa Calaborrano

et al., 2020. Generally, the maximum fraction of PANs in our budget (~50 %) agrees with Juncosa Calahorrano et al. (~40%) within our model uncertainties. Comparisons of particulate nitrate and organic nitrogen (gas or particulate) between our model run and the analysis of Calahorrano et al. are uncertain since our model does not account for gas-particle partitioning of nitrophenolics. Our model begins to deviate from the NO<sub>y</sub> budget trend seen by Calahorrano et al. once the sun sets, as expected.

#### 705 **3.4.2 NOx Lifetime**

The availability of O<sub>3</sub> and sunlight at emission strongly affects NO<sub>x</sub> lifetime ( $\tau_{NO_x}$ , Figure 11) defined below

$$\tau_{NO_X} = \frac{1}{\sum_i k_i} \tag{11}$$

where k<sub>i</sub> is a unimolecular rate coefficient for (R3, 8 – 10). Model runs with relatively large photolysis and O<sub>3</sub> at emission (Castle, Cow, and WF1) have near emission  $\tau_{NO_x}$  that range from 1 – 3 h (Figure 11), which are accompanied by larger total oxidation rates for all BBVOCs (SI Figure 15 – SI Figure 17). These model runs also exhibit the fastest nitrophenolic formation rates (section 3.3.2 and Figure 8). In contrast model runs with low or zero photolysis and near zero O<sub>3</sub> (WF2 and Dark) exhibit near emission  $\tau_{NO_x} = \sim 10 - 16$  h and  $\tau_{NO_x} = 20 - 150$  h, respectively. The absence of photolysis in the Dark model run explains the large difference in  $\tau_{NO_x}$  between the WF2 and Dark model runs as the WF2 model run has greater O<sub>3</sub> and P(NO<sub>3</sub>) that promotes NO<sub>3</sub> chemistry as well as OH radical that promotes PANs formation. In short, we find that "daytime" conditions have shorter NO<sub>x</sub> lifetimes, greater rates of BBVOC oxidation, and greater rates of nitrophenolics formation when compared

have shorter NO<sub>x</sub> lifetimes, greater rates of BBVOC oxidation, and greater rates of nitrophenolics formation when compared to "nighttime" conditions.

Once NO is depleted in both model runs NO<sub>x</sub> chemistry changes. The BBVOCs oxidation rate rapidly increases (SI Figure 15 – SI Figure 17) and NO<sub>x</sub> loss switches from primarily PAN and PNA to nitrophenolic production as the sun sets (Figure 10) and O<sub>3</sub> is entrained from the background. As such,  $\tau_{NO_x}$  decreases markedly to ~ 0.5 h.

- Due to their reduced oxidation rates at emission, the WF2 and Dark model runs retain about half (46% and 57 %, respectively) of the emitted NO<sub>x</sub> by sunrise the next day. Here, we calculate remaining NO<sub>x</sub> as the fraction of NO<sub>x</sub> remaining at the end of our model divided by the amount of NO<sub>x</sub> that was reacted, excluding dilution. This is about a  $\Delta NO_x/\Delta CO$  of ~4 ppbv ppmv<sup>-1</sup> at sunrise, which is similar to the initial emissions of Castle (~6 ppbv ppmv<sup>-1</sup>) and WF1 (~5 ppbv ppmv<sup>-1</sup>). Further, at sunrise, we expect the WF2 and Dark plumes to be more optically transparent and free of NO, and thus oxidation rates to increase
- rapidly as they both still contain NO<sub>x</sub>. An increase in oxidation at sunrise will likely be more important for the Dark model run, as it retains 61% of the emitted phenolics as opposed to 48 % in the WF2 model run. Plumes emitted after sunset have slower oxidation rates compared to daytime plumes (section 3.2), but undergo additional oxidation from evening to morning. However, outside of the plume-center, where O<sub>3</sub> is less effected by reaction with NO and is more likely to be generated by photochemical production, NO<sub>x</sub> loss rates may be much larger. Therefore, NO<sub>x</sub> away from the plume-center will likely be 730 depleted more rapidly.
- depleted more rapidi

#### **4** Conclusions

This study details the competitive oxidation of BBVOCs in four near-sunset, or low-photolysis, smoke plumes sampled by the NOAA Twin Otter or the NASA DC-8 aircraft during the FIREX-AQ 2019 field campaign. We model these plumes, as well as a theoretical dark plume, using an observationally constrained 0-D chemical box model.

Our key findings and arguments are summarized below.

#### • Section 2.4: Observations and Model Comparison

Our model achieves agreement with observed CO and O<sub>3</sub> typically within a difference of 10 %. However, strong O<sub>3</sub> gradients between plume center and edge can cause larger differences, specifically in the WF2 model run.

# Absolute differences between the model and observations of NO<sub>x</sub> and HONO are generally < 1 ppbv, but can be as large as 3.4 ppbv.</li>

- In most cases, BBVOC comparisons show that the model and observations agree within a factor of ~2, if not within observation errors.
- Model and observation agreement for phenolics and nitrophenolics is only available for two model runs (Castle and Cow) and most comparisons agree within observation errors, but some disagree by as much as a factor of 60.

#### • Section 3.1: Reactivity

- Our model suggests OH is reactive to most BBVOCs, while NO<sub>3</sub> is most reactive to phenolics, and O<sub>3</sub> to alkenes and terpenes.
- O Unlike urban plumes, NO<sub>3</sub> loss to NO, photolysis and heterogeneous uptake are negligible loss pathways.
   Most (≥97 %) of NO<sub>3</sub> loss occurs through BBVOC oxidation.
  - $\circ$  Reactivity of OH and O<sub>3</sub> is similar to, or greater than urban plumes, but NO<sub>3</sub> reactivity is a factor of 10 10<sup>4</sup> greater than typical urban plume reactivity.

#### • Section 3.2: Oxidation Rates

- Initial reactivity is a good indicator for subsequent oxidation by OH, but not for NO<sub>3</sub> and O<sub>3</sub>.
  - Phenolics are the only BBVOC group for which oxidation by NO<sub>3</sub>, OH, and O<sub>3</sub> is competitive.
  - $\circ$  The nitrate radical is responsible for 26 52 % of phenolic loss and leads (36%) phenolic oxidation in an optically thick mid-day plume.

#### Section 3.3: Phenolic Oxidation and Nitrophenolic Production

- All phenolic oxidation after sunset is dependent on  $O_3$ , whether through direct oxidation by  $O_3$ , production of NO<sub>3</sub> by NO<sub>2</sub> + O<sub>3</sub>, or ozonolysis of unsaturated hydrocarbons and subsequent decomposition to OH radicals.
  - Yields of nitrocatechol vary between 33 45 %.
  - Nitrate radical chemistry is responsible for 72 92 % (84 % in an optically thick mid-day plume) of nitrocatechol formation and controls nitrophenolic formation overall.

#### Section 3.4: Fate of NO<sub>x</sub> in Dark BB Plumes

- Formation of nitrophenolics by NO<sub>3</sub>, as opposed to OH, is the largest NO<sub>x</sub> sink and accounts for most of the inorganic and organic nitrogen at the end of the night.
- Nitrophenolic formation pathways account for 58 66 % of NO<sub>x</sub> loss by sunrise the following day.

• While both PANs and PNA account for most of the NO<sub>x</sub> loss shortly after emission, they decompose overnight providing a NO<sub>x</sub> source for nitrophenolic formation and increase nitrocatechol yield.

In short,  $NO_3$  chemistry should be considered, even during the daytime, when investigating BB plume oxidation as we find it is the main source of nitrophenolic formation in plumes studied here and thus may be a dominant pathway to SOA formation.

#### 775 Author Contributions

FIREX-AQ data were measured and processed by the following people: UW I<sup>-</sup> HR ToF CIMS (ZCJD, CDF, BBP, JAT); Tenax (KCB, PVR); Picarro G2401-m (MAR, SSB); NCAR CL (FMF, DDM, GST, AJW); UHSAS (AF, AMM); jNO<sub>2</sub> on the Twin Otter (MAR); CO by diode laser (JPD, GSD, HH, JBW); CO by CES (JP); NOAA CL (IB, JP, TBR); ACES (SSB, MAR, JL, RAW, CW); NOAA LIF (PSR, AWR); NOAA I<sup>-</sup> ToF CIMS (JAN, PRV); PAN (LGH, YRL); UIBK PTR ToF MS (FP, AW, GIG, KS, CS, MMC); SMPS/LAS (RHM, LT, El. Wi, Ed Wi); CAFS (SH, KU); smoke ages (CDH). Updates to the phenolic mechanism were performed by MMC, ZCJD, MAR, RHS. Model runs were conducted by ZCJD. Preparation of the manuscript was done by ZCJD with contributions from coauthors.

#### **Competing Interests**

The authors declare that they have no conflict of interest.

#### 785 Acknowledgements

Support of the UIBK PTR ToF MS came from Ionicon Analytik; Tomas Mikoviny provided technical assistance. Laura Tomsche and John Nowak supported the UIBK PTR ToF MS team as well. Thank you to Alan Fried, Dirk Richter, Jim Walega, and Petter Weibring for use of their HCHO measurements. A big thank you to all of those who helped organize and participated in the 2019 FIREX-AQ field campaign, specifically the NOAA Aircraft Operations, including Francisco Fuenmayor, Joe Greene, Conor Maginn, Rob Miletic, Joshua Rannenberg, and David Reymore.

#### **Financial Support**

Carley D. Fredrickson, Brett B. Palm, and Joel A. Thornton were supported by NOAA OAR Climate Program Office Award Number NA17OAR4310012. The UIBK PTR-ToF-MS instrument was partly funded by the Austrian Federal Ministry for

Transport, Innovation and Technology (bmvit) through the Austrian Space Applications Programme (ASAP) of the Austrian Research Promotion Agency (FFG). Felix Piel received funding from the European Union's Horizon 2020 research and

innovation program under grant agreement no. 674911 (IMPACT EU ITN). Zachary Decker received funding through a graduate research award from the Cooperative Institute for Research of Environmental Sciences.

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

#### 1135 doi:10.1029/2019EF001210, 2019.

Wolfe, G. M., Marvin, M. R., Roberts, S. J., Travis, K. R. and Liao, J.: The framework for 0-D atmospheric modeling (F0AM) v3.1, Geosci. Model Dev., 9(9), 3309–3319, doi:10.5194/gmd-9-3309-2016, 2016.

Xie, M., Chen, X., Hays, M. D., Lewandowski, M., Offenberg, J., Kleindienst, T. E. and Holder, A. L.: Light Absorption of Secondary Organic Aerosol: Composition and Contribution of Nitroaromatic Compounds, Environ. Sci. Technol., 51(20), 11607–11616, doi:10.1021/acs.est.7b03263, 2017.

Xing, J., Mathur, R., Pleim, J., Hogrefe, C., Gan, C. M., Wong, D. C., Wei, C., Gilliam, R. and Pouliot, G.: Observations and modeling of air quality trends over 1990-2010 across the Northern Hemisphere: China, the United States and Europe, Atmos. Chem. Phys., 15(5), 2723–2747, doi:10.5194/acp-15-2723-2015, 2015.

Yang, Y., Shao, M., Wang, X., Nölscher, A. C., Kessel, S., Guenther, A. and Williams, J.: Towards a quantitative

understanding of total OH reactivity: A review, Atmos. Environ., 134(2), 147–161, doi:10.1016/j.atmosenv.2016.03.010, 2016.

Yang, Y., Wang, Y., Zhou, P., Yao, D., Ji, D., Sun, J., Wang, Y., Zhao, S., Huang, W., Yang, S., Chen, D., Gao, W., Liu, Z., Hu, B., Zhang, R., Zeng, L., Ge, M., Petäjä, T., Kerminen, V. M., Kulmala, M. and Wang, Y.: Atmospheric reactivity and oxidation capacity during summer at a suburban site between Beijing and Tianjin, Atmos. Chem. Phys., 20(13), 8181–8200,

doi:10.5194/acp-20-8181-2020, 2020.

Yokelson, R. J., Andreae, M. O. and Akagi, S. K.: Pitfalls with the use of enhancement ratios or normalized excess mixing ratios measured in plumes to characterize pollution sources and aging, Atmos. Meas. Tech., 6(8), 2155–2158, doi:10.5194/amt-6-2155-2013, 2013.

El Zein, A., Coeur, C., Obeid, E., Lauraguais, A. and Fagniez, T.: Reaction Kinetics of Catechol (1,2-Benzenediol) and

1155 Guaiacol (2-Methoxyphenol) with Ozone, J. Phys. Chem. A, 119(26), 6759–6765, doi:10.1021/acs.jpca.5b00174, 2015.

## 1160Figures & Tables

### Table 1: Details of fires studied.

| Fire name | County/State     | Latitude | Longitude | Date    | Time sampled    | Aircraft | Fuel             |
|-----------|------------------|----------|-----------|---------|-----------------|----------|------------------|
|           |                  |          |           | sampled |                 |          |                  |
| Williams  | Ferry/Washington | 47.9392  | -118.6183 | Aug 07  | 16:30–17:45 PDT | DC-8     | Short grass,     |
| Flats     |                  |          |           |         | & 18:00–19:30   |          | ponderosa timber |
|           |                  |          |           |         | PDT             |          |                  |
| Castle    | Coconino/Arizona | 36.5312  | -112.2281 | Aug 21  | 18:00-19:15     | Twin     | Mixed conifer    |
|           |                  |          |           |         | MST             | Otter    |                  |
| 204 Cow   | Grant/Oregon     | 44.2851  | -118.4598 | Aug 28  | 20:00-22:00 PDT | Twin     | Primarily        |
|           |                  |          |           |         |                 | Otter    | lodgepole pine   |
|           |                  |          |           |         |                 |          | with conifer     |