# Peer review of "Nighttime and Daytime Dark Oxidation Chemistry in Wildfire Plumes: An Observation and Model Analysis of FIREX-AQ Aircraft Data"

_Atmospheric Chemistry and Physics, 2021_

## Referee Comment (RC1)

**Summary/recommendations**

This study combines FIREX-AQ biomass burning (BB) observations of dusk and nighttime plumes with a chemistry box model to elicit numerous details related to $NO_3$, $O_3$, and OH oxidation, BBVOC oxidation, and product formation in the plumes. Nighttime and/or optically thick plume chemistry remains understudied in the active BB world. This study did an excellent job detailing anticipated chemistry, with an emphasis on $NO_3$ and $NO_x$ reactions. It was well-written and easy to read. I especially commend the authors on their admirable constraint with regards to sentence length. The figures are generally quite extensive but are mostly presented in an attractive, consistent, clear, and readable manner. While readable and clear, the paper is quite long and the authors may want to consider whether any details could be moved to the supplement. I recommend publication after addressing my mostly minor comments below.

**General comments**

Line 139: I would like a little more discussion on how close to or far away from Lagrangian sampling each flight was. If this has been well characterized, perhaps an SI table is appropriate. 'Semi-Lagrangian', while often used in the BB literature, is vague.

Sect 2.1: It's no secret that UHSAS have struggled with saturation in high-aerosol environments such as wildfire plumes. While not one of the most crucial measurements in this paper, I recommend including a brief discussion in the text or SI about its performance for the specific plumes used in this work.

Related, the UHSAS (Twin Otter) and SMPS (DC-8) are on 1 Hz vs 60 s timescales. How did you account for this difference in your analysis? (Especially since you assumed center-line modeling and the SMPS almost certainly did not capture only the center of a plume in 60 s).

Section 2.1.1-2.1.2: (Twin Otter and NASA DC-8 descriptions) Were there any opportunities during FIREX to characterize instruments (specifically, instruments measuring the same species) from each aircraft against each other? If yes or no, could you briefly detail. Is it anticipated that 'real' individual differences in a given species present at each fire studied greatly outweigh differences (uncertainty) between two different instruments and platforms measuring said species?

Line 202: Why only one "Dark" case? Please provide a brief justification, including why you chose the WF2 case as the dark case's template.

Lines 236-7: what qualifies as "small" uptake coefficients and aerosol diameters here? Are the aerosol diameters appropriate for this equation (3) appropriate for the (non-coarse) mode/s observed?

Some of the discussion between Sect 2.3.1 Chemistry and Emissions and Sects 2.3.2-3 seem a little disconnected. Are you using the ER inventory (Sect 2.3.1) only for compounds not measured directly that are still important to MCM mechanisms? Also it seems that Eq 1 does not care about background (out-of-plume) concentrations of species. Is that the case, and if so can you briefly justify using total rather than background-corrected concentrations (mixing ratios)?

Relatedly, it's slightly unclear to me exactly what the box model used consists of. It uses the MCM and a NOAA F0AM BB mechanism. Are those the only components to it (along with the dilution rate discussed)? Has this exact model been used elsewhere?

Line 320: make sure to note that you are using 'delta' notation to denote background-corrected (I don't think this notation was previously defined).

Line 326-329: briefly justify why you're using the latter transects of the WF2 fire rather than the former.

Line 381 & elsewhere: Why is formaldehyde separated from other BBVOCs in the analysis?

Lines 409-411: Why do BB plume have more pronounced reactivity for NO3 than for OH or O3? If discussed elsewhere, point to that discussion.

Line 496: Akherati et al., 2020 also an appropriate citation here:
Oxygenated Aromatic Compounds are Important Precursors of Secondary Organic Aerosol in Biomass-Burning Emissions: Ali Akherati, Yicong He, Matthew M. Coggon, Abigail R. Koss, Anna L. Hodshire, Kanako Sekimoto, Carsten Warneke, Joost de Gouw, Lindsay Yee, John H. Seinfeld, Timothy B. Onasch, Scott C. Herndon, Walter B. Knighton, Christopher D. Cappa, Michael J. Kleeman, Christopher Y. Lim, Jesse H. Kroll, Jeffrey R. Pierce, and Shantanu H. Jathar, Environmental Science & Technology 2020 54 (14), 8568-8579, DOI: 10.1021/acs.est.0c01345

Line 588-593 (paragraph) be careful with wording here. For example- "Further, the nitrocatechol yield changes to 27 % – 50 % (Figure 9 D) when varying total BBVOC emissions by a factor 0.5 – 4." Change the order of one of these number pairs--from Fig 9D the yield is 50% at a BBVOC factor of 0.5 and drops to 27% by a BBVOC factor of 4.

Lines 606-609 (paragraph): If the reactions & temperature dependence are uncertain, how did you obtain an estimate of phenolics/NO2 for 268 K?

Line 640-41: citation for these estimated PNA and PAN lifetimes?

Line 643 & associated figure caption: define CH3O2NO2 (methyl peroxy nitrate?).

**Figures/Tables**
Figure 2: while useful for Sect 2 discussion, I suggest considering whether this figure could be moved to the SI. It is quite large and there are already an extensive number of detailed figures.

Figure 3 caption: I suggest reminding the reader what 'all model runs' means in "Average (all model runs).." .

Figure 4: I suggest increasing whitespace between the OH and NO3 bar clusters just a bit to make the distinction between the two more clear.

Figure 5: same comment, please space the 'subpanels' out a little more (that is, add more whitespace in between each subplot).

Figure 7: note for final publication that the right-hand line on panel D got cut off. I suggest adding a legend for the different colors (O3, NO3, OH) within the figure to make it more easily interpretable to the reader (rather than having to repeatedly refer to the figure caption). Could copy over legend from Fig 6.

Figure 8: it's a little difficult to read 'Castle'-- could consider enlarging all fire names on this.

Figure 9: Panels C and D appear to be out of order (not going from left to right) . Also 'BBVOC' is used throughout in the text-either update figure captions from VOC or note in caption that you are using 'VOC' for "BBVOC" here, if that's correct. Or note why you use VOC instead of BBVOC here if it's for another reason.

**Supporting information**
I'm fine with having multiple short tables on the same page, but I suggest increasing the whitespace between each paper. Since the title and caption are in the same font, page 11 of the SI is hard for me to quickly parse. Also, isn't it ACP convention to put table captions at the top of the table? (Or is that just what most people have done previously.)

Page S2, Mechanism section (line 1202)-I'm a little confused by the comment "reactions in red are already in the MCM and will need to be replaced when used in conjunction with an MCM mechanism." Maybe the meaning here is clear to regular MCM users, but by "replace" do you mean "remove"? Could you expand on this note a little more to make the meaning more clear?

**Technical comments**

Line 186: looks like the WF1 and WF2 fires should be referred to as Fig 1B and 1C. Note that in discussing the Cow fire (lines 195-199) you don't refer to Fig 1E.

Line 376: switches between X and x, are these meant to be consistent (X or x)?

Line 424: missing parentheses around Decker citation.

---

## Author Comment (AC1)

**Responses to Referee 1 Comments**

1. Line 139: I would like a little more discussion on how close to or far away from Lagrangian sampling each flight was. If this has been well characterized, perhaps an SI table is appropriate. 'Semi-Lagrangian', while often used in the BB literature, is vague.

We thank the reviewer for the suggestion. We have added details on the emission time of the plume center smoke for each fire plume sampled. Emission time is calculated by subtracting the estimated smoke age from the sample time. The spacing of Twin Otter plume intercepts were close to Lagrangian sampling. The difference of emission times of smoke sampled by the Twin Otter at plume center is < 10 min. The difference of emission times of smoke sampled by the DC-8 at plume center is between 30 and 60 min. The differences between each aircraft are due to air speed. The Twin Otter flies at ~70 m s-1 compared to the DC-8 air speed of ~170 m s-1.

We added SI Table 2, shown below, along with additional text to the manuscript.

| Transect | WF 1 (Aug 7 2019)               | WF2                           | Castle (Aug 22 2019)    | Cow (Aug 29 2019)               |
|----------|---------------------------------|-------------------------------|-------------------------|---------------------------------|
| 1        | $23:01:04 \pm 5.0 \text{ min}$  | Aug 8 00:36:01 ± 8.0 min      | $01:01:41 \pm 1.2 \min$ | $01:30:59 \pm 71.5 \text{ min}$ |
| 2        | 22:46:13 ± 6.6 min              | Aug 8 00:18:18 ± 7.7 min      | $00:59:58 \pm 1.4 \min$ | $01:27:45 \pm 68.1 \text{ min}$ |
| 3        | 22:43:11 ± 3.8 min              | Aug 8 00:09:45 ± 6.2 min      | $00:59:55 \pm 1.7 \min$ | $01:30:34\pm55.6~\mathrm{min}$  |
| 4        | 22:33:25 ± 8.6 min              | Aug 7 23:53:59 ± 7.2 min      | $00:52:11 \pm 3.8 \min$ |                                 |
| 5        | 22:13:04 ± 13.7 min             | Aug 7 23:29:05 $\pm$ 12.8 min |                         |                                 |
| 6        | 21:58:06 ± 12.8 min             | Aug 7 23:24:59 $\pm$ 8.2 min  |                         |                                 |
| 7        | 21:51:34 ± 16.5 min             | Aug 7 23:14:38 $\pm$ 6.2 min  |                         |                                 |
| 8        | 21:37:17 ± 15.6 min             | Aug 7 22:50:45 ± 11.4 min     |                         |                                 |
| 9        | 21:13:38 ± 19.9 min             | Aug 7 22:41:39 ± 22.5 min     |                         |                                 |
| 10       | $20:55:25 \pm 30.2 \text{ min}$ |                               |                         |                                 |
|          |                                 |                               |                         |                                 |

Estimated Emission Time at Plume Center (UTC)  $\pm$  uncertainty (min)

SI Table 1: List of estimated emission times (UTC) with uncertainty (min) for each plume. Emission times for transects used to constrain the model are bolded.

Added to section 2.1.1

"Even so, estimated emission times (calculated from estimated plume ages) suggest smoke sampled on successive intercepts at the Castle and Cow plume centers were emitted within 3- and 10-min time periods, respectively. However, plume age uncertainties for the Cow plume are large (SI Table 1)."

Added to section 2.1.2

"However, smoke emission times for the plume center of WF1 and WF2 covered a larger time period ( $\sim$ 30 – 60 min) compared to the NOAA Twin Otter (SI Table 1)."

2. Sect 2.1: It's no secret that UHSAS have struggled with saturation in high-aerosol environments such as wildfire plumes. While not one of the most crucial measurements in this paper, I recommend including a brief discussion in the text or SI about its performance for the specific plumes used in this work.

Indeed, saturation can be an issue for the UHSAS, however we were able to correct for this in flight. The following text was added to the manuscript in section 2.1.1.

"The UHSAS data were corrected for coincidence up to a factor to 1.4, following the method described in Kupc et al 2018. The sample for the UHSAS was diluted up to a factor 2.9 for part of the flights to increase accuracy at higher concentrations."

3. Related, the UHSAS (Twin Otter) and SMPS (DC-8) are on 1 Hz vs 60 s timescales. How did you account for this difference in your analysis? (Especially since you assumed center-line modeling and the SMPS almost certainly did not capture only the center of a plume in 60 s).

To account for the lower time resolution of the SMPS, we took a conservative approach by using the maximum observed aerosol surface area. Related to this, we have realized that the SMPS does not capture larger aerosol (>  $\sim$ 250 nm) and therefore we have instead used the Laser Aerosol Spectrometer (LAS) to calculates surface area. Doing so has resulted in a change in surface area of a factor of up to 100 in the WF1 and WF2 box model runs. Even so, the resulting N2O5 heterogeneous reactivity accounts for <  $\sim$ 2.5% of total NO3 and N2O5 combined reactivity. The increase is small because of the substantial NO3 reactivity found in the wildfire plumes we study.

In response to this change, we have updated SI Figure 11 and related text. The cases for  $\gamma_{NO_3} = 0.1$  and  $\gamma_{NO_3} = 1$  were removed from the figures because they are unrealistically large NO3 uptake coefficients.

4. Section 2.1.1-2.1.2: (Twin Otter and NASA DC-8 descriptions) Were there any opportunities during FIREX to characterize instruments (specifically, instruments measuring the same species) from each aircraft against each other? If yes or no, could you briefly detail. Is it anticipated that 'real' individual differences in a given species present at each fire studied greatly outweigh differences (uncertainty) between two different instruments and platforms measuring said species?

The Twin Otter and NASA DC-8 were based at the same airfield for a period of time during FIREX-AQ, but they unfortunately did not execute coordinated flights sampling the same fires on the same day to provide direct comparison between the different sets of instruments.

Among the relevant instruments that could have been directly compared, the stated uncertainties are < 3 % for CO, < 10 % for NO and NO2, <3 % for O3, and < 10 % for  $j_{NO2}$ . Calibrations for HONO have larger uncertainties (30 % for the UW I- CIMS and 15 % + 3 pptv for the NOAA I- CIMS). As shown in Figure 2 (which includes the above uncertainties), the calibration and stated measurement uncertainty is typically smaller than any differences between transects, and certainly smaller than differences across sampled fire plume. Therefore we anticipate that the actual individual differences in a given species present at each fire studied greatly outweigh differences (uncertainty) between two different instruments and platforms measuring said species.

5. *Line 202: Why only one "Dark" case? Please provide a brief justification, including why you chose the WF2 case as the dark case's template.*

We chose only one "Dark" case as to not lengthen the analysis and paper further than it already is. Further "Dark" model runs were outside the scope for this analysis and therefore were not included. The WF2 case was chosen as a template for the "Dark" case because the smoke was emitted near sunset (unlike WF1), provided sufficient emissions such that chemistry would continue throughout the night (unlike Castle), and the observations were not already in the dark (unlike Cow).

In response, the first paragraph of section 2.3 was modified as shown below.

"We modeled smoke plumes from three fires (Castle, Cow, and WF). We present four model cases (Castle, Cow, WF1, WF2) constrained by aircraft observations and one case (Dark) identical to the WF2 case except all modeled photolysis frequencies are set to zero. We consider the dark model run only for the WF2 case and not the others since it is a hypothetical exercise intended to illustrate the evolution of smoke emitted after dark, a case for which there were no available observations from the 2019 campaign. The Dark case simulates oxidation of the WF2 plume if it was emitted after sunset. The Dark case is used to understand the effect of photolysis on the WF2 model run."

6. Lines 236-7: what qualifies as "small" uptake coefficients and aerosol diameters here? Are the aerosol diameters appropriate for this equation (3) appropriate for the (non-coarse) mode/s observed?

We thank the reviewer for the questions. We have added the following text to the manuscript to clarify.

"For large particle diameters or large uptake coefficients, the simplified heterogeneous uptake equation requires a correction for gas phase diffusion (Fuchs & Sutugin, 1970; Kolb et al., 2010). For accumulation mode particles of order 100 nm and uptake coefficients of order 0.01, this correction is not important."

7. Some of the discussion between Sect 2.3.1 Chemistry and Emissions and Sects 2.3.2-3 seem a little disconnected. Are you using the ER inventory (Sect 2.3.1) only for compounds not measured directly that are still important to MCM mechanisms? Also it seems that Eq 1 does not care about background (out-of-plume) concentrations of species. Is that the case, and if so can you briefly justify using total rather than background-corrected concentrations (mixing ratios)?

We thank the reviewer for the comment.

For clarity, we have modified the text in section 2.3.1 to inform the reader that further details of how emission ratios are used will be given in section 2.3.3, shown below.

"We initiate the model, as discussed in section 2.3.3., using an emissions inventory of 302 BBVOCs in the form of emission ratios (ERs)."

The reviewer's question about our use of the ER inventory for model initiation is answered by the two excerpts from section 2.3.3 below.

"In all plumes except the Castle plume, our first transect sampled smoke 36  $\min - 2$  h old and therefore we implemented an iterative method (McDuffie et al., 2018; Wagner et al., 2013) to estimate initial emissions (at age = 0). We began with best-guess estimates of CO, NO, NO2, HONO and O3 then systematically changed these initial conditions to minimize the differences between model output and observations downwind."

"Initial conditions in the Castle run were taken directly from observations of NO, NO2, O3, CO, HONO, phenol, catechol, cresol, and methylcatechol in the first transect where the plume age was  $3 \pm 1$  min, and therefore was close

to age = 0. We initiated the remaining 298 BBVOCs by using CO and Eq. (1)."

However, these excerpts are separated by text describing the iterative method. We have modified the manuscript by combining the above paragraphs at the beginning of section 2.3.3 and making clarifying additions, as shown below.

"In all plumes except the Castle plume, our first transect sampled smoke 36 min – 2 h old and therefore we implemented an iterative method (McDuffie et al., 2018; Wagner et al., 2013) to estimate initial emissions (at age = 0). We began with best-guess estimates of CO, NO, NO2, HONO, O3 and all BBVOCs (determined by CO and our emissions inventory by Eq. (1)) then systematically changed these initial conditions to minimize the differences between model output and observations downwind. Initial conditions in the Castle run were taken directly from observations of NO, NO2, O3, CO, HONO, phenol, catechol, cresol, and methylcatechol in the first transect where the plume age was  $3 \pm 1$  min, and therefore was close to age = 0. We initiated the remaining 298 BBVOCs by using CO and Eq. (1). Initial conditions for all cases are shown in SI Table 5."

It is correct that Eq. (1) does *not* include background (out-of-plume) concentrations of species. This is by definition since emission ratios describe an initial (age=0) emission relative to another emission, in this case CO. In other words, emission ratios are specific only to a fuel type and are independent of background conditions. Alternatively, Normalized Excess Mixing Ratios (Eq. 5) account for background conditions and are used to describe observations when age>0. To clarify this point, the following text was added immediately after Eq. (1).

"Note that an ER is used to describe an emission (when smoke age = 0) and is different than a Normalized Excess Mixing Ratio (defined in section 2.4.1) used to describe observations when smoke age >0."

8. Relatedly, it's slightly unclear to me exactly what the box model used consists of. It uses the MCM and a NOAA FOAM BB mechanism. Are those the only components to it (along with the dilution rate discussed)? Has this exact model been used elsewhere?

All of the components of this model have been used in other applications, but their combination is specific to this paper. To clarify this point to the reader, we have added the following text to the end of section 2.3 Box Model Description

"Components of our model have been used for other applications (Decker et al., 2019; McDuffie et al., 2018; Robinson et al., 2021; Wagner et al., 2013). However, the combination of the components is specific to this work." 9. Line 320: make sure to note that you are using 'delta' notation to denote background-corrected (I don't think this notation was previously defined).

The following text was added

"(where  $\Delta$  indicates background-corrected)"

10. Line 326-329: briefly justify why you're using the latter transects of the WF2 fire rather than the former.

We use the latter transects of the WF2 Fire plume because only the later transects show a monotonic decrease in CO (most easily seen in SI Figure 3), which was one criterion we used for selection observations to be used in the model. This is described in section 2.3.2 shown below.

"We chose transects that showed a monotonic decrease of CO with distance from the fire, cover an age range of at least one hour, and have similar emission times as shown in SI Figure 2 - 3 and SI Table 1."

We have edited the lines in question to direct the reader to section 2.3.2, as shown below.

"In order to avoid these changes, we use only observations from the latter to constrain our model, as discussed in section 2.3.2."

11. Line 381 & elsewhere: Why is formaldehyde separated from other BBVOCs in the analysis?

Formaldehyde is a simple aldehyde and therefore does not fall into one of the general BBVOC categories shown in the top pane of Figure 3 (i.e. furans, terpenes, etc.). However, it has a substantial contribution to OH reactivity. Therefore, we specify HCHO separately. We have added the following text to clarify.

"In this analysis we do not specify an aldehyde group, and therefore separate HCHO from the general BBVOC groupings."

12. Lines 409-411: Why do BB plume have more pronounced reactivity for NO3 than for OH or O3? If discussed elsewhere, point to that discussion.

We thank the reviewer for this suggestion. The increased reactivity of NO3 is due to the specific emissions from biomass burning, such as oxygenated aromatics and furans that have substantial reactivity towards NO3. The concept of increased NO3 reactivity in BB plumes is introduced in section 1 Introduction and copied below.

"This is the result of elevated concentrations of several highly reactive BBVOCs within the plume. Specifically, directly emitted aromatic alcohols (phenolics) react with NO3 at near the gas-kinetic limit to form nitrophenolics, a subset of nitroaromatics, and secondary organic aerosol (Finewax et al., 2018; Lauraguais et al., 2014; Liu et al., 2019; Xie et al., 2017)." To emphasize this point, we have added the following sentence after the lines in question.

"The increased reactivity of NO3 is due to the specific emissions from biomass burning, such as phenolics and furans that have substantial reactivity towards NO3. The compounds greatly increase NO3 reactivity compared to urban VOC profiles, but do not increase OH reactivity to the same degree."

 Line 496: Akherati et al., 2020 also an appropriate citation here: Oxygenated Aromatic Compounds are Important Precursors of Secondary Organic Aerosol in Biomass-Burning Emissions: Ali Akherati, Yicong He, Matthew M. Coggon, Abigail R. Koss, Anna L. Hodshire, Kanako Sekimoto, Carsten Warneke, Joost de Gouw, Lindsay Yee, John H. Seinfeld, Timothy B. Onasch, Scott C. Herndon, Walter B. Knighton, Christopher D. Cappa, Michael J. Kleeman, Christopher Y. Lim, Jesse H. Kroll, Jeffrey R. Pierce, and Shantanu H. Jathar, Environmental Science & Technology 2020 54 (14), 8568-8579, DOI: 10.1021/acs.est.0c01345

We thank the reviewer for their suggestion. The reference has been added.

14. Line 588-593 (paragraph) be careful with wording here. For example- "Further, the nitrocatechol yield changes to 27 % – 50 % (Figure 9 D) when varying total BBVOC emissions by a factor 0.5 – 4." Change the order of one of these number pairs--from Fig 9D the yield is 50% at a BBVOC factor of 0.5 and drops to 27% by a BBVOC factor of 4.

We thank the reviewer for their suggestion. The following text was altered

Original: "Further, the nitrocatechol yield changes to 27 % - 50 % (Figure 9 D) when varying total BBVOC emissions by a factor 0.5 - 4."

Revision: "Further, the nitrocatechol yield changes to 27 % - 50 % (Figure 9 D) when varying total BBVOC emissions from 4 to 0.5."

15. Lines 606-609 (paragraph): If the reactions & temperature dependence are uncertain, how did you obtain an estimate of phenolics/NO2 for 268 K?

The estimate at 268 K is determined by using the  $k_{NO_2+O_3}$  rate coefficient at 268 K, but using the  $k_{O_3}$  rate coefficient at 298 K. The text was altered, as shown below, to clarify.

**Original:**

For example, the above calculated ratio could be much lower in cold lofted plumes, but knowledge of temperature dependent  $O_3$  + phenolic rate coefficients are unavailable. Using temperatures observed in the WF2 plume (~268 K) for  $k_{NO_2+O_3}$  the phenolics to NO2 ratio at which NO3 and O3 oxidation is equal would be ~ 4.

**Revision:**

For example, the above calculated ratio could be much lower in cold lofted plumes, but knowledge of temperature dependent  $O_3$  + phenolic rate coefficients ( $k_{0_3}$ ) are unavailable. Using temperatures observed in the WF2 plume (~268 K) for  $k_{NO_2+O_3}$  (but using  $k_{0_3}$  at 298 K) the phenolics to NO2 ratio at which NO3 and O3 oxidation is equal would be ~ 4.

16. Line 640-41: citation for these estimated PNA and PAN lifetimes?

The PAN and PNA lifetimes are determined from the model directly. The temperature dependent rate coefficients for each are taken from IUPAC recommended rates (Atkinson et al., 2006). This citation is now included in the main text, as shown below.

"The WF2 plume is lofted, and therefore cold (~267 K), which results in a long PAN and PNA lifetime (~150 h, and ~0.4 h, respectively, calculated from the model directly (Atkinson et al., 2006))."

17. Line 643 & associated figure caption: define CH3O2NO2 (methyl peroxy nitrate?).

This is now defined in the main text as shown below.

"The increase in NO2 after sunset promotes methyl peroxy nitrate (CH3O2NO2) as well as NO3 chemistry products, which grow steadily overnight."

**Figures/Tables**

18. Figure 2: while useful for Sect 2 discussion, I suggest considering whether this figure could be moved to the SI. It is quite large and there are already an extensive number of detailed figures.

We thank the reviewer for the suggestion. After careful consideration, we have decided to keep Figure 2 in the main text. Comments 1 and 2 from reviewer 2 are concerned with a lack of model and observation comparisons as well as making the comparisons clear to the reader. To address these comments from reviewer 2, we have decided to keep Figure 2 because it is the only model and observation comparison presented in the main text.

19. Figure 3 caption: I suggest reminding the reader what 'all model runs' means in 'Average (all model runs)..".

We thank the reviewer for the suggestion. The text within the parentheticals was altered as follows

Original: (of all model runs)

**Revision: (of all five model runs)**

20. Figure 4: I suggest increasing whitespace between the OH and NO3 bar clusters just a bit to make the distinction between the two more clear.

We thank the reviewer for the suggestion. Additional white space was added. Also, the y-axis of  $O_3$  and  $NO_3$  was modified to match the y-axis of OH. The Original and revised figures are below.

Original: